# STXD: Structural and Temporal Cross-Modal Distillation for Multi-View 3D Object Detection

**Sujin Jang**[*1]  **Dae Ung Jo**[*1]  **Sung Ju Hwang**[2]  **Dongwook Lee**[1]  **Daehyun Ji**[1]

Samsung Advanced Institute of Technology (SAIT) [1]
Korea Advanced Institute of Science and Technology (KAIST) [2]
{s.steve.jang, daeung.jo, dw12.lee, derek.ji}@samsung.com
sjhwang82@kaist.ac.kr

## Abstract

3D object detection (3DOD) from multi-view images is an economically appealing alternative to expensive LiDAR-based detectors, but also an extremely challenging task due to the absence of precise spatial cues. Recent studies have leveraged the teacher-student paradigm for cross-modal distillation, where a strong LiDAR-modality teacher transfers useful knowledge to a multi-view-based image-modality student. However, prior approaches have only focused on minimizing global distances between cross-modal features, which may lead to suboptimal knowledge distillation results. Based on these insights, we propose a novel structural and temporal cross-modal knowledge distillation (STXD) framework for multi-view 3DOD. First, STXD reduces redundancy of the feature components of the student by regularizing the cross-correlation of cross-modal features, while maximizing their similarities. Second, to effectively transfer temporal knowledge, STXD encodes temporal relations of features across a sequence of frames via similarity maps. Lastly, STXD also adopts a response distillation method to further enhance the quality of knowledge distillation at the output-level. Our extensive experiments demonstrate that STXD significantly improves the NDS and mAP of the based student detectors by $2.8\% \sim 4.5\%$ on the nuScenes testing dataset.

## 1 Introduction

3D object detection (**3DOD**) is the task of locating and classifying objects in 3D space using input data from specific modalities, which typically include LiDAR point clouds [61, 14, 7] and camera images [54, 34, 24]. 3DOD is central to understanding the surrounding environment and has been widely applied to various complex vision systems, such as autonomous driving [44], robotic manipulation [71, 65], and augmented reality [42]. Recently, camera-based multi-view 3DOD has emerged as an attractive alternative to expensive LiDAR-based methods, thanks to its ubiquity, low cost, and the semantically rich information from colorized pixels. In general, current works on multi-view 3DOD aim to extract a unified bird's-eye view (**BEV**) feature map from multiple camera images. However, 3DOD from only camera-specific BEV features is extremely challenging due to the lack of precise spatial cues [39], which leads to inferior performance of camera-based models compared to LiDAR-based models. To bridge the gap, some prior works [16, 11, 19, 29, 9] introduced a teacher-student paradigm for cross-modal knowledge distillation to transfer geometrically-rich 3D information from LiDAR models (**teacher**) to camera-based 3DOD models (**student**). The current state-of-the-art 3DOD models typically follow a similar detection paradigm, where modality-specific encoders are used to extract BEV features, followed by a 3D object detection head [29, 34, 55, 7].

---

[*]These authors equally contributed to this work

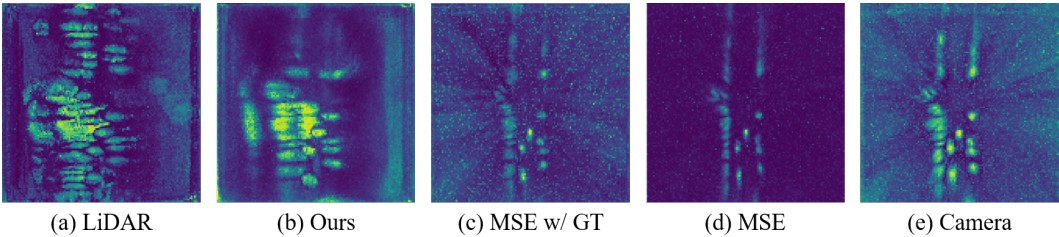

| (a) LiDAR | (b) Ours | (c) MSE w/ GT | (d) MSE | (e) Camera |

Figure 1: Examples of BEV features generated by **(a)** the LiDAR model, **(b)-(d)** the model trained with different feature-level distillation methods, and **(e)** the camera-only model. Here we visualize the $\mathcal{L}_2$-norm value of the BEV features. More examples are provided in the Appendix.

Hence, recent works have taken the BEV features from different modalities as the focal point of knowledge distillation [16, 29], and then applied additional response-level distillation [11, 19, 9].

One of the main challenges in cross-modal knowledge distillation is transferring modality-specific knowledge from the teacher to the student in a different modality. This is primarily because the features learned by different modalities are typically non-homogeneous, and there exist distributional divergences between the feature spaces as shown in Fig.1-(a) and (e). Most existing distillation methods aim to minimize global distances between cross-modal features (*i.e.,* $\mathcal{L}_2$-distance). While this approach has achieved some success, it fails to capture the structural knowledge inherent in the modality-specific features, which can lead to suboptimal knowledge distillation, as shown in Fig.1-(c) and (d). To address these challenges, we first propose a distillation method that regularizes cross-correlation of BEV features from the teacher and the student models. In particular, we introduce a decorrelation mechanism [67, 2] to reduce redundancy among feature components, thereby maximizing the information contained in the features. Our structural knowledge distillation approach ultimately prevents information collapse in the student model and enhances the component-wise feature similarities across different modalities, as illustrated in Fig.1-(a) and (b).

We also explore the integration of temporal knowledge embedded in features of previous frames. Although temporal information has been actively exploited in recent 3DOD approaches [34, 29, 32, 41], it has been relatively understudied for cross-modal distillation. The main challenge is the spatio-temporal misalignment between past and current frames. For instance, in driving scenes, foreground objects may have different locations or disappear over time. Thus, directly matching the misaligned cross-modal features could result in incorrect distillation. Instead, we leverage the temporal relations of the BEV features by generating temporal similarity maps between the teacher's current and past frames. Specifically, we train the student model to generate the BEV feature of the current frame, which mimics the temporal relations encoded in the similarity maps from the teacher. In this way, we can avoid false matching issues when transferring temporal information over a sequence of frames. To further improve the quality of transferred knowledge, we also adopt the response distillation approach [10, 59, 19, 9], which assigns quality scores to the predictions of the teacher model, thereby enabling the distillation of only meaningful information at the prediction-level.

Our structural and temporal cross-modal distillation (**STXD**) framework integrates the proposed feature- and response-level distillation approaches (see Fig.2), and demonstrated a significant improvement on the nuScenes 3DOD benchmark [3], leading to up to 3.2% NDS and 4.5% mAP improvement over the based student models. We also conducted detailed ablation studies to validate the effectiveness of each component of STXD. Notably, our framework does not require additional computational cost at testing time, while improving the prediction performance of the student models.

## 2 Related Work

**LiDAR-based 3DOD.** Recent LiDAR-based 3DOD detectors can be categorized based on the representation types of 3D space: voxel-based and point-based approaches. The voxel-based approaches [69, 58, 27, 48, 14, 7] convert point clouds into a regular voxel-grid representation. Point-based approaches [62, 49, 64, 61] extract point-wise features directly from raw point clouds. Both approaches involve transforming low-level features into a BEV representation, which is then followed by an object detection head that generates response features.

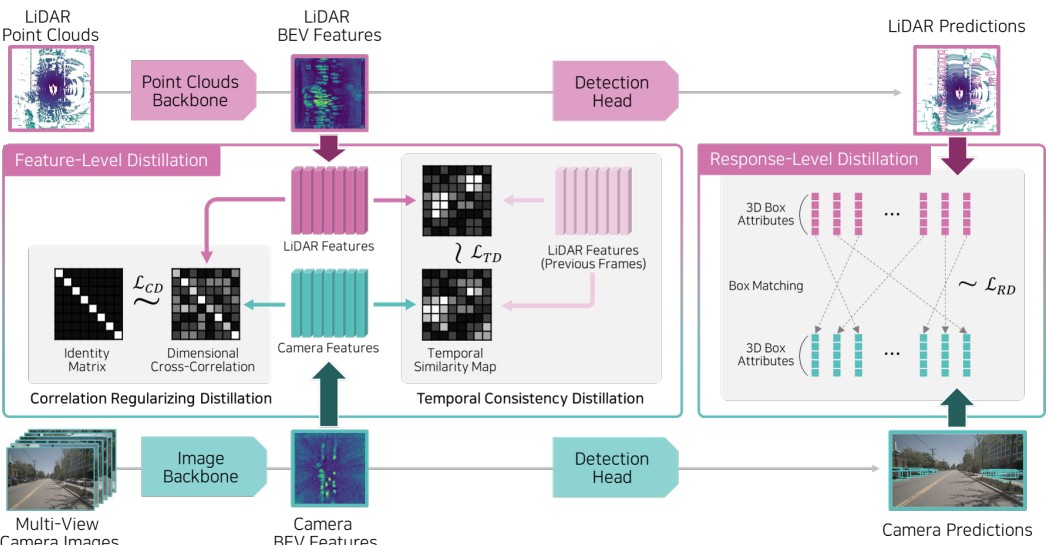

Figure 2: An overview of STXD framework for multi-view 3D object detection. Structural and temporal feature-level distillation is applied to the unified bird's-eye view (BEV) features extracted and transformed from both LiDAR- and camera-based backbones (Sec.3.2 and 3.3). Response-level distillation is applied to the 3D box candidates predicted by both LiDAR- and camera-based detectors (Sec.3.4). During testing, only the camera model is utilized without additional computational cost.

**Camera-based 3DOD.** Camera-based approaches typically perform 3DOD on either single- or multi-view images. Prior works on single-view 3DOD aim to detect 3D boxes directly from image features [5, 38, 53], or using intermediate geometric representations (*e.g.,* pseudo-lidar [66, 12, 50], probabilistic depth-cues [47, 52]). More recently multi-view 3DOD approaches [21, 33, 29] use the lift-splat-shoot method [43, 47] to transform multi-image features into a unified bird's-eye view (BEV) representation followed by an object detection head. Other works on multi-view images [54, 34, 35, 24] aim to implicitly learn query-based BEV features sampled from corresponding 2D image features based on attention mechanisms [4, 72]. While some of these works [50, 47, 33] utilize LiDAR point clouds as input for explicit supervision of depth cues, they do not fully exploit the potential of LiDAR-based 3DOD features, which have appeared to be effective for improved prediction performance of camera-based models. In light of this work, we further provide new perspectives on the structural and temporal knowledge embedded in the LiDAR-based BEV features.

**Multi-Modal 3DOD.** Recently, integrating complementary information from different modality sensors, especially LiDAR and cameras, has demonstrated improved 3DOD performance. One line of work fuses intermediate feature representations in the BEV space [8, 31, 36, 63], while other approaches try to fuse object proposals from each modality in the detection head [25, 6, 1]. Another line of work augments point clouds with semantic information extracted from corresponding images [51, 51, 57]. However, these methods require both sensors to be available during test time. In contrast, our cross-modal distillation approach leverages both modalities only during training to transfer useful knowledge from LiDAR-based to camera-based 3DOD models. During testing, we use only multi-camera images without any additional computational costs.

**Knowledge Distillation for 3DOD.** Most prior works on knowledge distillation for object detection have focused on transferring knowledge among single-modal detectors, particularly for 2D object detection (*e.g.,* [15, 59]) and LiDAR-based models (*e.g.,* [60, 56]). Relatively fewer works have studied cross-modal distillation for object detection. MonoDistill [11] projects LiDAR point clouds into perspective image space to distill spatially aligned LiDAR features to a camera-based student model. LIGA-Stereo [16] aims to transfer intermediate features from LiDAR to a stereo camera-based student model. CMKD [19] and BEVDistill [9] employ knowledge distillation from LiDAR-based models at both the feature and response levels to minimize the gap between modalities. Similarly, UVTR [29] tries to directly match the BEV representations of the camera model to the LiDAR-based

teacher. Although these methods have achieved some success, they still do not carefully consider the distributional divergences between non-homogeneous modalities. To mitigate the modality gap between LiDAR and camera features, TIG-BEV [22] learns the inner geometry of foreground objects by encouraging the student's features to replicate the self-correlation of the teacher's features. However, this method still does not consider cross-modal associations among BEV features from the teacher and student models. Differs from previous methods, our distillation approach considers structural knowledge in the features by introducing cross-modal cross-correlation regularization inspired by the decorrelation mechanism [67, 2]. Furthermore, we introduce a method for effectively distilling temporal information contained in previous frames of the teacher. We also utilize response-level distillation to further improve the quality of knowledge distillation at the predictions level.

## 3 Methodology

### 3.1 Preliminary

Recent 3D object detectors share a similar detection paradigm, where low-level features are extracted from modality-specific encoders and subsequently transformed into BEV features (see Fig.2). For example, modern LiDAR-based detectors [69, 27, 55] use 3D encoders to extract low-level features from the raw point clouds, then the features are further transformed to the BEV features $\mathbf{F} \in \mathbb{R}^{X \times Y \times Z \times D}$ with spatial extent of $X \times Y \times Z$ and the number of channels $D$. For recent multi-view image detectors, there are two mainstream approaches for transforming the low-level image features to the BEV features: (1) estimating depth cues [21, 33, 29] via lift-splat-shoot [43] and (2) implicit learning of depth via cross-attention mechanism [35, 34]. Regardless of the type of transform methods, the BEV features can be represented by $\mathbf{G} \in \mathbb{R}^{X \times Y \times Z \times D}$, similar to the LiDAR-based detectors. In this work, as illustrated in Fig.2, we aim to train a high-performing camera-based detector (student) by distilling useful knowledge from a LiDAR-based detector (teacher) while considering the structural and temporal information embedded in the modality-specific BEV featuers, $\mathbf{F}$ and $\mathbf{G}$.

### 3.2 Correlation Regularizing Distillation

Prior works on cross-modal distillation directly force the student to mimic the BEV feature from the teacher by calculating element-wise distance loss (*i.e.,* $||\mathbf{F} - \mathbf{G}||_2$, [19, 29]), or weighted via foreground mask (*i.e.,* $||\mathbf{W}_{fg} \cdot (\mathbf{F} - \mathbf{G})||_2$, [11, 9]). However, they ignore the structural knowledge inherent in modality-specific features and may not effectively distill rich information contained in the features. Recently, several self-supervised learning methods have incorporated cross-correlation regularization to maximize the information contained in features [67, 2]. These methods can also be applied to the problem of LiDAR-to-camera knowledge distillation, since LiDAR and camera features can be regarded as different representations obtained from the same scene, and have a spatial one-to-one correspondence. From this intuition, we propose the **C**orrelation Regularizing **D**istillation (**CD**) loss that maximizes the similarity between aligned features while regularizing cross-correlation along feature dimensions. Let batches of $N$ aligned $D$-dimensional LiDAR and camera features be given as $\mathbf{F} = [\mathbf{f}_1, \mathbf{f}_2, ..., \mathbf{f}_N]^T \in \mathbb{R}^{N \times D}$ and $\mathbf{G} = [\mathbf{g}_1, \mathbf{g}_2, ..., \mathbf{g}_N]^T \in \mathbb{R}^{N \times D}$, respectively. $N$ is the number of serialized BEV features, given by $X \cdot Y \cdot Z$. Following [67], each feature is transformed to be mean-centered along the batch dimension, denoted by $\hat{\mathbf{F}}$ and $\hat{\mathbf{G}}$. Then, CD loss is defined as:

$$\mathbf{C} = \hat{\mathbf{F}}^T \hat{\mathbf{G}} \in \mathbb{R}^{D \times D}, \tag{1}$$

$$\mathcal{L}_{CD} := \sum_i (1 - \mathbf{C}(i,i))^2 + \lambda_c \sum_i \sum_{j \neq i} \mathbf{C}(i,j)^2, \tag{2}$$

where $\mathbf{C}$ denotes a dimensional cross-correlation matrix and $\lambda_c \geq 0$ is a balancing hyper-parameter. The first term of Eq.2 encourages the aligned LiDAR and camera feature components to be similar, while the second term reduces the correlation among feature components that are not aligned. Hence, CD loss can reduce duplicated information in the learned features, allowing for the student to learn diverse information from the teacher. We analyzed the effectiveness of the CD loss in Sec.4.2.

### 3.3 Temporal Consistency Distillation

In driving scenes, sensor data is received sequentially. Therefore, the past few frames may contain valuable information for detecting objects in the current frame, and incorporating temporal information has brought significant improvements to 3DOD [34, 29, 32, 41]. However, directly learning

from the LiDAR features of the past frames (*i.e.,* via $\mathcal{L}_2$-distance) may not be effective due to the spatially and temporally misaligned non-homogeneous features. To address this issue, we propose the **T**emporal Consistency **D**istillation (**TD**) loss, which indirectly distills the teacher's information from past frames by introducing a temporal similarity map. Let batches of aligned LiDAR and camera features of the current frame be given as $\mathbf{F}^{(0)}$ and $\mathbf{G}^{(0)}$, respectively. We are also given the LiDAR features of the $k$-th previous frame, denoted by $\mathbf{F}^{(-k)}, k \in [1, K]$, where $K$ is the maximum number of past frames referred to. Then, the cross-modal temporal similarity map $\mathbf{S}^{(-k)}$ and the intra-modal temporal similarity map $\mathbf{T}^{(-k)}$ between the current and the $k$-th previous frame are calculated as:

$$\mathbf{S}^{(-k)} = \mathbf{G}^{(0)}\mathbf{F}^{(-k)T} \in \mathbb{R}^{N \times N}, \quad \mathbf{T}^{(-k)} = \mathbf{F}^{(0)}\mathbf{F}^{(-k)T} \in \mathbb{R}^{N \times N}, \quad k \in [1, K]. \tag{3}$$

Then TD loss is defined as:

$$\mathcal{L}_{TD} := \sum_k D_{KL}(\mathbf{S}^{(-k)}||\mathbf{T}^{(-k)}), \tag{4}$$

where $D_{KL}$ is the Kullback-Leibler divergence (**KLD**). Since $\mathbf{S}^{(-k)}$ and $\mathbf{T}^{(-k)}$ are matrices, we apply a softmax function to each row of them, and then calculate the KLD for each row. Based on the temporal similarity map, TD loss distills the teacher's information of past frames indirectly, avoiding spatial false matching issues across the time frames. We also demonstrated and analyzed the effectiveness of TD in Sec.4.2.

### 3.4 Response-Level Distillation

In general, detector models infer a large number of candidates, but not all candidates contain meaningful information. Since only a few candidates detect target objects, it is necessary for a student to selectively learn from the predictions of the teacher. Inspired by recent works on prediction-guided distillation [10, 59, 19, 9], we define the **R**esponse-Level **D**istillation (**RD**) loss that assigns quality scores to the predictions of the teacher model, enabling the distillation of only meaningful information. Let us define $\{(\mathbf{b}_i, \mathbf{c}_i)\}_{i=1}^M$ as prediction results of teacher model, where $\mathbf{b}_i$ and $\mathbf{c}_i$ are regression and classification results for the $i$-th of $M$ box candidates, respectively. In a similar manner, student prediction can be defined as $\{(\tilde{\mathbf{b}}_j, \tilde{\mathbf{c}}_j)\}_{j=1}^{\tilde{M}}$. Following [10, 59], quality score $q_i$ for $i$-th candidate of the teacher predictions can be calculated as follow:

$$q_i = \left(c_i^*\right)^{1-\gamma} \cdot \left(\text{IoU}\left(\mathbf{b}_i^*, \mathbf{b}_i\right)\right)^\gamma, \tag{5}$$

where $c_i^*$ is the predicted probability for the class of the ground truth box that is matched to $i$-th candidate of the teacher predictions, $\mathbf{b}_i^*$ is a regression parameter of the ground truth box that is matched to $i$-th candidate of the teacher predictions, and $\gamma$ is a hyper-parameter balancing classification and localization. Then RD loss is defined as:

$$\mathcal{L}_{RD} := \sum_j q_{\pi(j)} \cdot \left(\|\mathbf{b}_{\pi(j)} - \tilde{\mathbf{b}}_j\|_1 + D_{KL}(\mathbf{c}_{\pi(j)}||\tilde{\mathbf{c}}_j)\right), \tag{6}$$

where $\pi(j)$ is a function that returns the index of the corresponding teacher's prediction matched with the $j$-th student's prediction. By combining RD with our structural and temporal distillation (CD,TD), we achieved additional improvement on the detection performance as reported in Sec.4.2.

### 3.5 Optimization Objective

Finally, the overall optimization objective for the student models can be obtained by integrating the aforementioned three distillation losses and the task loss $\mathcal{L}_{task}$:

$$\mathcal{L}_{total} = \mathcal{L}_{CD} + \mathcal{L}_{TD} + \mathcal{L}_{RD} + \mathcal{L}_{task}, \tag{7}$$

where $\mathcal{L}_{task}$ is adopted from a general set-to-set target assignment for the student with ground truth, including regression and classification losses [54], based on the Hungarian algorithm [26].

## 4 Experiments

We now present extensive experimental results to validate our structural and temporal cross-modal knowledge distillation framework (STXD) for 3D object detection (3DOD) task. We first introduce the details of experimental setting (Sec.4.1) followed by various ablation studies to validate the effectiveness of each component in STXD (Sec.4.2). Then, we compare STXD against several leading methods on the nuScenes [3] dataset (Sec.4.3). Additional experiments are provided in the Appendix.

## 4.1 Experimental Settings

**Dataset.** We evaluate our STXD framework on the nuScenes [3] benchmark dataset, which is large-scale and widely used by recent research works on 3D object detection. The nuScenes dataset comprises 1,000 driving scenes with a duration of approximately 20s, which are divided into 700, 150, and 150 scenes for training, validation, and testing, respectively. Each scene consists of RGB images collected by six cameras, which capture a $360°$ field of view at 12Hz and 32-beam LiDAR point clouds at 20Hz. Annotations for 23 categories of objects are provided for each sample at a rate of 2Hz, and the metrics for the 3DOD task are evaluated using 10 classes of objects, as defined by the official evaluation metrics. These metrics include mean Average Precision (mAP) and nuScenes Detection Score (NDS). In detail, mAP measures the recall and precision of predicted bounding boxes, while NDS is a weighted sum of mAP and five true positive nuScenes metrics: mean Average Translation Error (mATE), mean Average Scale Error (mASE), mean Average Orientation Error (mAOE), mean Average Velocity Error (mAVE), and mean Average Attribute Error (mAAE).

**Baseline Models.** To validate the effectiveness of our approach, we adopt two state-of-the-art cross-modal distillation approaches as baselines. Similar to concurrent work [9], we use BEVFormer [34] as the image-modality student model and Object-DGCNN [55] as the LiDAR-modality teacher model, where the DGCNN attention module of the transformer-decoder is replaced with vanilla multi-head attention. For the second baseline, we adopt the multi-view camera (single frame of UVTR-C, multi-frame of UVTR-CS) and LiDAR (UVTR-L) models from UVTR [29] as the student and teacher models, respectively. Our base student models represent two mainstream methods for transforming 2D perspective image features into 3D BEV features, where BEVFormer uses cross-attention mechanism [4, 72] and UVTR-C/CS is based on estimating depth distributions through lift-splat-shoot approach [43, 47].

**Implementation Details.** The input image size for the student models is set to $1600 \times 900$. For both teacher and student models, we use a voxel size of $(0.1m, 0.1m, 0.2m)$, and the input point clouds are filtered within a range of $[-51.2m, 51.2m]$ for the X and Y axes and $[-5.0m, 3.0m]$ for the Z axis. The dimensions of the BEV feature grids are $128 \times 128$, and the number of channel $D$ for BEV features is set to 256. We train all student models for 24 epochs without using CBGS [70]. $\lambda_c$ in Eq.2 is set to 0.01 according to [67]. More implementation details are provided in the Appendix.

## 4.2 Ablation Studies

For our ablation studies, we implemented BEVFormer [34] with ResNet-101-DCN as backbone network for the student model, and evaluated the trained model on the nuScenes validation dataset.

**Ablation on Distillation Methods.** To validate the effects of each proposed distillation approach in Sec.3, we conducted an ablation study on the loss function. As shown in Tab.1, each loss term contributes to the improvement in performance. Specifically, the proposed feature-level distillation (CD and TD) plays a crucial role in enhancing the performance. When CD and TD losses are applied solely, NDS is increased by 2.47% and 1.77%, respectively. When all loss terms are combined, the performance is further improved without any conflicts, resulting in the best performance with a significant improvement of NDS by 2.87% and mAP by 3.52%. These results demonstrate the effectiveness of our structural and temporal distillation as well as the response distillation method.

Table 1: Ablation on our distillation losses.

| CD | TD | RD | NDS(%) | mAP(%) |
|----|----|----|--------|--------|
| -  | -  | -  | 51.44  | 40.51  |
| -  | -  | ✓  | 52.88  | 42.07  |
| -  | ✓  | -  | 53.21  | 42.66  |
| ✓  | -  | -  | 53.91  | 43.26  |
| -  | ✓  | ✓  | 53.70  | 42.86  |
| ✓  | -  | ✓  | 53.89  | 43.42  |
| ✓  | ✓  | -  | 54.00  | 43.41  |
| ✓  | ✓  | ✓  | **54.31** | **44.03** |

Table 2: Comparisons of feature-level distillation methods. Bottom results are obtained by applying different representation learning strategies to CD loss.

| Method | NDS(%) | mAP(%) |
|--------|--------|--------|
| w/o KD | 51.44 | 40.51 |
| MSE | 51.67 | 41.01 |
| MSE w/ GT | 51.84 | 41.03 |
| CD w/ VICReg [2] | 52.04 | 40.89 |
| CD w/ CLIP [45] | 52.47 | 41.69 |
| CD w/ Barlow Twins [67] | **53.91** | **43.26** |

**Comparison with Feature-Level Distillation Methods.** To solely examine the effectiveness of our correlation regularizing distillation (CD), we compared it to the following feature-level distillation methods without applying other distillation approaches. MSE is a baseline method [19, 29] that simply minimizes the element-wise $\mathcal{L}_2$-distance between the features of a teacher and a student. MSE w/ GT is similar to MSE, but it only distills features near the ground truth point [11, 9]. Our CD loss in Eq.2 draws inspiration from the Barlow Twins [67] approach for representation learning. To validate our choice, we also conducted a comparison with other alternative methods. Specifically, we applied VICReg [2] and CLIP [45] to CD loss and evaluated their impact.

Tab.2 highlights that even in the absence of other distillation losses, the CD loss alone delivers the most promising performance, resulting in a significant increase of $2.47\%$ in NDS. At the bottom of Tab.2, we observe that other representation learning methods also lead to performance improvements. Notably, the Barlow Twins method achieves the best performance among the tested methods. From a distillation perspective, VICReg solely relies on MSE loss to transfer the teacher's knowledge, which is referred to as the invariance term in [2]. As a result, the structural knowledge from the teacher may not be fully exploited, leading to suboptimal performance. CLIP is a contrastive learning method that typically requires a large and diverse set of positive and negative pairs of features for effective learning. In the case of 3DOD, however, cross-modal feature batches extracted from a scene typically contain only a limited number of positive pairs, while the number of negative pairs is much larger. Consequently, the effectiveness of contrastive learning may be compromised in the context of 3DOD. In contrast, Barlow Twins leverages the structural knowledge by exploiting cross-correlation between the cross-modal features (see Eq.1), and is robust to the size of feature set [67]. Thus, our correlation regularizing distillation (CD) is the most effective when implemented with Barlow Twins.

**Correlation Regularization Effect of CD Loss.** In Sec.3.2, we discussed that distillation with correlation regularization aims to maximize the information contained in the learned features, allowing the individual components of the feature to become more informative. Here we validate our argument using two qualitative measures: (1) effective dimension and (2) dimensional redundancy.

**(1) Effective dimension** [46] quantitatively measures the dimensionality of features. A higher effective dimension indicates lower redundancy among the features. In Tab.3, we compare effective dimensions $d_{\text{eff}}$ of features learned through various feature-level distillation methods. Since $d_{\text{eff}}$ is bounded by $\sqrt{D} = 16$, we also provide the squared values for a intuitive interpretation, bounded by $D = 256$. For example, the features learned through CD can be effectively represented in a dimensional space of at least 20 dimensions, approximately. Notably, our CD loss achieves the highest effective dimension compared to other feature distillation methods.

Table 3: Effective dimension $d_{\text{eff}}$ for different feature-level distillation methods and teacher.

| Method | $d_{\text{eff}}$ | $d_{\text{eff}}^2$ |
|---|---|---|
| MSE | 3.810 | 14.519 |
| MSE w/ GT | 4.059 | 16.474 |
| **CD** (Ours) | **4.389** | **19.267** |
| Teacher | 5.757 | 33.137 |

**(2) Dimensional redundancy:** Redundant feature components exhibit higher correlation among themselves. From this intuition, we quantify the redundancy in features by utilizing Eq.1 and 2. For camera features $\mathbf{G}$, we define a dimensional self-correlation matrix $\mathbf{C}_{\text{self}} = \hat{\mathbf{G}}^T \hat{\mathbf{G}}$. We then calculate the second term of Eq.2 for $\mathbf{C}_{\text{self}}$ and refer to the resulting value as the dimensional redundancy of $\mathbf{G}$. Fig.3 depicts the dimensional redundancy of features learned from the MSE and CD loss throughout the training progress. Results demonstrate that the CD loss is more effective in reducing dimensional redundancy compared to the MSE loss. Furthermore, even when the MSE loss is replaced with the CD loss during training, a decrease in dimensional redundancy is still observed (red dashed lines).

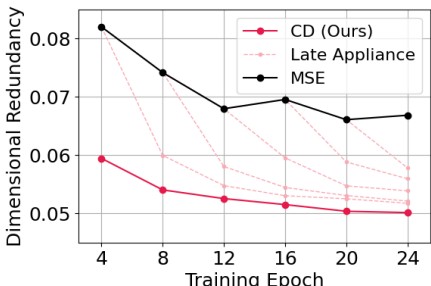

Figure 3: Trend of dimensional redundancy of features over training progress of the student model.

These results indicate that distilling features through a simple element-wise comparison can lead to dimensional collapse. In contrast, our information maximization distillation method leverages the structural knowledge embedded in the feature space by considering the cross-correlation of feature components, and ultimately contributes to significantly improved prediction performance (see Tab.1).

Table 4: Performance comparisons on the nuScenes validation set. For fair comparisons, we use ResNet-101-DCN as backbone for all the experiments. † indicates our implementation of BEV-Former [34] with modified BEV grid size of 128×128. ∗ indicates our implementation of Object-DGCNN [34] with a modified transformer-decoder module. In modality column, 'L', 'C', and 'L → C' represent LiDAR, Camera, and knowledge distillation from LiDAR to Camera, respectively.

| Method | Modality | NDS(%) | mAP(%) | mATE↓ | mASE↓ | mAOE↓ | mAVE↓ | mAAE↓ |
|---|---|---|---|---|---|---|---|---|
| Obj-DGCNN∗ [55] | L | 66.7 | 60.9 | 0.332 | 0.260 | 0.307 | 0.264 | 0.202 |
| UVTR-L [29] | L | 66.4 | 59.3 | 0.345 | 0.259 | 0.313 | 0.218 | 0.185 |
| DETR3D [54] | C | 42.5 | 34.6 | 0.773 | 0.268 | 0.383 | 0.842 | 0.216 |
| PETR [35] | C | 44.2 | 37.0 | 0.711 | 0.267 | 0.383 | 0.865 | 0.201 |
| PolarFormer [24] | C | 52.8 | 43.2 | 0.648 | 0.270 | 0.348 | 0.409 | 0.201 |
| BEVDepth [33] | C | 53.8 | 41.8 | - | - | - | - | - |
| BEVFormer† [34] | C | 51.4 | 40.5 | 0.690 | 0.275 | 0.360 | 0.362 | 0.194 |
| +BEVDistill [9] | L → C | 52.4 | 41.7 | - | - | - | - | - |
| +STXD (Ours) | L → C | **54.3** | **44.0** | 0.635 | 0.264 | 0.365 | 0.309 | 0.198 |
| UVTR-C [29] | C | 44.1 | 36.2 | 0.758 | 0.272 | 0.410 | 0.758 | 0.203 |
| +L2C [29] | L → C | 45.0 | 37.2 | 0.735 | 0.269 | 0.397 | 0.761 | 0.193 |
| +STXD (Ours) | L → C | **46.1** | **39.0** | 0.698 | 0.265 | 0.413 | 0.755 | 0.203 |
| UVTR-CS [29] | C | 48.3 | 37.9 | 0.731 | 0.267 | 0.350 | 0.510 | 0.200 |
| +L2CS [29] | L → C | 48.8 | 39.2 | 0.720 | 0.268 | 0.354 | 0.534 | 0.206 |
| +STXD (Ours) | L → C | **50.8** | **41.4** | 0.662 | 0.265 | 0.381 | 0.484 | 0.199 |

**Effect of $K$ on TD Loss.** The temporal consistency distillation (TD) in Eq.4 utilizes the BEV features extracted from the past $K$ frames. Tab.5 demonstrates increasing the value of $K$ results in improved 3DOD performance, particularly in terms of mAVE (mean Average Velocity Error). By leveraging the information from consecutive frames through the TD loss, the student can effectively learn the dynamics of target objects, leading to accurate predictions of their dynamic attributes. This highlights the impor-

Table 5: Performance of the proposed method depending on $K$.

| $K$ | NDS(%) | mAP(%) | mAVE↓ |
|---|---|---|---|
| 1 | 54.06 | 43.48 | 0.3246 |
| 2 | 54.13 | 43.45 | 0.3148 |
| 3 | **54.31** | **44.03** | **0.3093** |

tance of referencing past frames, as it allows the teacher to transfer a greater amount of information to the student, resulting in improved 3DOD performance. Similar results have also been echoed in concurrent works [32, 41], which also leverage temporal information for the single-modal 3DOD.

**Visualization of Temporal Similarity Map.** The similarity map defined in Eq.3 calculates the element-wise similarity between the BEV features of the current and past frames. In Fig.4, we select the feature element corresponding to the location of the *car* object in the current BEV space and calculate its similarity to the entire features of the past frames. Notably, the student model trained with TD exhibits improved similarity patterns with the teacher (see Fig.4-(a) and (b)). This indicates that the student model has learned to capture similar temporal relations and representations

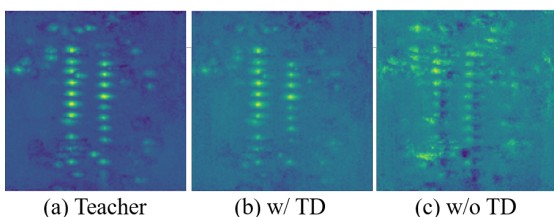

(a) Teacher    (b) w/ TD    (c) w/o TD

Figure 4: Examples of temporal similarity maps for (a) teacher model, (b) student model trained with TD, and (c) student model trained without TD.

from the teacher by referring to the past frames. The competence of our temporal distillation approach ultimately contributes to the improved 3DOD performance, as demonstrated in our ablation studies (see Tab.1 and 5). We provide more temporal similarity visualization results in the Appendix.

## 4.3 Comparisons with Other Methods

To demonstrate the generalizability of our approach, we apply STXD to both UVTR [29] and BEVFormer [34] baselines. For a fair comparison, we use ResNet-101-DCN [18, 13] as the backbone of our student models, unless otherwise specified. Tab.4 shows the evaluation results on the nuScenes

Table 6: Performance comparisons on the nuScenes testing set. † indicates implementation of BEV-Former [34] with modified BEV grid size of 128×128. ‡ indicates the use of V2-99 [28] as backbone, otherwise ResNet-101-DCN is used for a fair comparison. In modality column, 'L', 'C', and 'L → C' represent LiDAR, Camera, and knowledge distillation from LiDAR to Camera, respectively.

| Method | Modality | NDS(%) | mAP(%) | mATE↓ | mASE↓ | mAOE↓ | mAVE↓ | mAAE↓ |
|---|---|---|---|---|---|---|---|---|
| DETR3D [54] | C | 47.9 | 41.2 | 0.641 | 0.255 | 0.394 | 0.845 | 0.133 |
| PETR [33] | C | 45.5 | 39.1 | 0.647 | 0.251 | 0.433 | 0.933 | 0.143 |
| PolarFormer [24] | C | 54.3 | 45.7 | 0.612 | 0.257 | 0.392 | 0.467 | 0.129 |
| BEVFormer† [34] | C | 52.6 | 42.4 | 0.620 | 0.263 | 0.457 | 0.387 | 0.132 |
| +STXD (Ours) | L → C | **55.5** | **46.5** | 0.579 | 0.259 | 0.462 | 0.363 | 0.113 |
| BEVFormer‡ [34] | C | 55.5 | 45.7 | 0.605 | 0.263 | 0.371 | 0.375 | 0.116 |
| +STXD (Ours) | L → C | **58.3** | **49.7** | 0.546 | 0.256 | 0.383 | 0.356 | 0.117 |
| UVTR-C [29] | C | 43.0 | 36.4 | 0.724 | 0.266 | 0.486 | 0.898 | 0.143 |
| +L2C [29] | L → C | 44.0 | 38.2 | 0.677 | 0.262 | 0.493 | 0.925 | 0.150 |
| +STXD (Ours) | L → C | **45.8** | **40.2** | 0.634 | 0.271 | 0.465 | 0.918 | 0.143 |
| UVTR-CS [29] | C | 48.6 | 39.0 | 0.702 | 0.263 | 0.435 | 0.548 | 0.144 |
| +L2CS [29] | L → C | 48.7 | 39.8 | 0.679 | 0.262 | 0.432 | 0.612 | 0.133 |
| +STXD (Ours) | L → C | **51.8** | **43.5** | 0.614 | 0.260 | 0.407 | 0.589 | 0.132 |

validation set, where STXD significantly improves the performance of the based student models, surpassing the performance of existing distillation methods [29, 9], up to $2.9\%$ in NDS and $3.5\%$ in mAP. In most cases, the addition of STXD contributes to the improvement of mATE (translation), mASE (scale), and mAVE (velocity) metrics. Both mATE and mASE are closely related to the geometric features of 3D objects, and the teacher effectively transfers such important information to the student models. The improvement in mAVE can be attributed to our temporal distillation method.

Similar results are echoed in our experiments on the nuScenes testing dataset, as shown in Tab. 6. We did not use CBGS [70] or test time augmentations to evaluate the sole effect of our method. STXD consistently improves NDS and mAP by up to $3.2\%$ and $4.5\%$, respectively, compared to the various baseline methods (UVTR-C/CS, L2C/L2CS, BEVFormer†). STXD also improves performances for the student model with heavy backbone (BEVFormer‡) by $2.8\%$ in NDS and $4.0\%$ in mAP. These results are significant as models with heavy backbones already achieve high performance, making it challenging to further improve through knowledge distillation. In the Appendix, we show the competence of STXD on lightweight backbones as well. These results demonstrate the effectiveness and versatility of STXD in enabling cross-modal distillation between the teacher and the student models. Overall, STXD enables the student to learn the maximum amount of information from the teacher, including geometric and temporal knowledge, leading to significantly enhanced performance.

## 5 Conclusion

In this paper, we presented STXD, a cross-modal knowledge distillation framework for the multi-view 3D object detection task. STXD is capable of effectively distilling structural and temporal knowledge across different modalities through Correlation Regularizing Distillation (CD) and Temporal Consistency Distillation (TD). To further enhance the distillation quality, STXD also adopts Response-Level Distillation (RD), which transfers task-specific knowledge at the output level. Our extensive experiments and ablation studies demonstrated the effectiveness of STXD on the nuScenes dataset, where the NDS and mAP of the based student detectors are improved by up to $3.2\%$ and $4.5\%$ on the testing set. Inspired by recent works on multi-modal fusion [36, 63, 1] and other types of 3D perception tasks (*e.g.,* BEV Segmentation [68, 20], Occupancy Detection [23, 30]), we plan to further explore the potential of STXD in such various tasks leveraging cross-modal distillation. It would also be valuable to apply and validate our approach with other models that utilize long-term information such as [17, 41]. We hope our work provides a solid baseline and new perspectives on structural and temporal cross-modal distillation.

**Societal Impact.** Though the proposed method significantly improves the performance of multi-view camera-based 3D object detection, it does not guarantee perfect predictions for all cases. Therefore, when applied to actual autonomous driving scenes, a contingency plan must be included.

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

# Appendix

In this appendix, we provide additional details of the implementation and training of our method (Appendix A). We also present additional experimental results, including ablations on lightweight backbones, additional baselines, and qualitative examples (Appendix B). These results further demonstrate the effectiveness of our STXD framework in cross-modal distillation for 3D object detection.

## A   Additional Implementation Details

### A.1   Details on Cross-Modal Distillation Baselines

In Sec.4.1 of the manuscript, we introduced two cross-modal distillation baselines for 3D object detection: (1) BEVFormer [34] as the student model and Object-DGCNN [55] as the teacher model; and (2) UVTR-C/CS as the student models and UVTR-L as the teacher model [29]. To be more specific, in the first baseline (Object-DGCNN → BEVFormer), we set the spatial dimensions of the bird's-eye view (BEV) feature to be $X = Y = 128$, and $Z = 1$ for both the teacher and student models, indicating a flat BEV feature space. In accordance with the concurrent work [9], we replaced the DGCNN attention module of the transformer-decoder in the teacher model with vanilla multi-head attention. Apart from these modifications, we followed the original implementations of both the student and teacher models. In the second baseline (UVTR-L → UVTR-C/CS), we utilized the original implementation of UVTR [29] without any modifications. The dimensions of the bird's-eye view (BEV) features were set to $X = Y = 128$ and $Z = 11$ for UVTR-C/CS; and $Z = 5$ for UVTR-L. Here, $Z$ represents the height of the Bird's Eye View (BEV) feature space. The BEV features were further sampled using the $(x, y, z)$ positions of object queries $Q$, where $|Q| = 900$.

Note that we adopted two baseline student models that utilize multi-frame inputs. BEVFormer generates the BEV feature of the current frame by referring to the BEV feature of the past frames via temporal self-attention mechanism. In UVTR-CS, multi-frame images are concatenated and forwarded to the image encoder.

### A.2   Training Details

For a fair comparison, we used ResNet-101-DCN pretrained by FCOS3D [53] as the backbone network for the student models in our extensive experiments (see Sec.4.2 and 4.3 in the manuscript) following prior works [34, 29]. We also showcased the competence of our STXD framework with other types of backbones, including VoVNet-99 [28] initialized from DD3D [40] as well as lightweight ResNet-18 and ResNet-50 networks (see B.1). We set the balancing parameter $\gamma$ to 0.8 and 0.6 for classification and localization in Eq.5 of the manuscript, following [10, 59]. The weights for each loss term in Eq.7 of the manuscript are set to $(0.1, 100.0, 1.0, 1.0)$, respectively, in the order mentioned. The student models were trained for 24 epochs using a learning rate of $2 \times 10^{-4}$, and a batch size of 1 per GPU. All models were trained on 8 of NVIDIA A100 GPU while following the original codebase for each model. We employed AdamW [37] as the optimizer with a weight decay of $1 \times 10^{-2}$. To isolate the effect of our STXD framework in cross-modal distillation, we deliberately did not use the CBGS strategy [70] or test-time augmentation in our experiments.

### A.3   Instance Matching Function in Response-Level Distillation

In Sec.3.4, we adopted the commonly used Hungarian algorithm [26] to define set-to-set matching between the ground-truth and candidates from the teacher and the student, separately. Then, we constructed a mapping function $\pi(j)$ between candidates from the teacher and the student based on matched ground-truth indices.

# B Additional Experimental Results

## B.1 Ablations on Lightweight Backbones

To further validate the effectiveness of our STXD framework in practical scenarios where lighter backbone networks are preferred for deployment on edge devices, we conducted experiments using smaller backbone networks for the student model. This allows us to assess the performance of our approach under resource-constrained settings while improving the level of accuracy in 3D object detection. We trained BEVFormer with ResNet-18 and ResNet-50 backbone networks on nuScenes training set and evaluated on the validation set. The results in Tab.7 demonstrate that STXD significantly improves the performance of BEVFormer with smaller backbone networks where it achieves an improvement of 3.24% and 4.00% in NDS and 3.33% and 4.49% in mAP, respectively. These improvements are consistent with the results presented in Tab.4 and 6 of the manuscript, further confirming the versatility and compatibility of our approach across different backbone architectures.

Table 7: Performance of STXD on lightweight backbone networks for the student model.

| Method | Backbone | NDS(%) | mAP(%) |
|---|---|---|---|
| BEVFormer [34] | ResNet-18 | 40.42 | 28.70 |
| +**STXD** (Ours) | ResNet-18 | **43.66** | **32.03** |
| BEVFormer [34] | ResNet-50 | 44.06 | 32.86 |
| +**STXD** (Ours) | ResNet-50 | **48.06** | **37.35** |

## B.2 Experiments on Additional Teacher-Student Baseline

To further validate the effectiveness of our approach across various teacher-student baselines, we extend our experiments to additional student baseline, BEVDepth [33], which relies on LSS-based feature generation [43], depth supervision, and multi-frame inputs. Here we followed the official implementation of TiG-BEV [22] for cross-modal distillation with CenterPoint [64] as the teacher model, but added our feature-level distillation losses for fair comparisons. Results are reported in Tab.8, where our approach outperforms prior distillation methods, such as BEVDistill [9] and TiG-BEV [22], by up to +3.2% of NDS. These results are consistent to the results from BEVFormer and UVTR-C/CS baselines in Tab.4 and 6.

Table 8: Performance comparison of distillation methods on the baseline with CenterPoint as the teacher and BEVDepth as the student. TiG-BEV (FD): applying only the feature-level distillation (FD) losses ($\mathcal{L}_{\text{inter-channel}}$, $\mathcal{L}_{\text{inter-keypoint}}$). TiG-BEV: leveraging full set of losses including inner-depth supervision proposed in TiG-BEV. STXD (CD): applying our correlation regularizing distillation loss. STXD (TD): applying our temporal consistency distillation loss.

| Method | Backbone | Modality | NDS(%) | mAP(%) |
|---|---|---|---|---|
| CenterPoint [64] | VoxelNet | L | 64.6 | 56.4 |
| BEVDepth [33] | ResNet-50 | C | 43.1 | 32.9 |
| +BEVDistill [9] | ResNet-50 | L $\rightarrow$ C | 45.4 | 33.2 |
| +TiG-BEV (FD) [22] | ResNet-50 | L $\rightarrow$ C | 45.2 | 35.8 |
| +TiG-BEV [22] | ResNet-50 | L $\rightarrow$ C | 46.1 | 36.6 |
| +**STXD (CD)** | ResNet-50 | L $\rightarrow$ C | **48.4** | **37.1** |
| +**STXD (TD)** | ResNet-50 | L $\rightarrow$ C | **48.3** | **36.1** |

## B.3 Comparison to Correlation-based Distillation Method

Our Correlation Regularizing Distillation (CD) and TiG-BEV [22] both consider dimensional correlation of features. However, these methods are distinct in their underlying objectives and implementations. TiG-BEV proposes the inter-channel BEV distillation loss (IC loss) that encourages the student's features to replicate the dimensional self-correlation of the teacher's features (*i.e.,* minimize $||\mathbf{F}^T\mathbf{F} - \mathbf{G}^T\mathbf{G}||_2$ ). In contrast, the core contribution of CD is introducing cross-modal

cross-correlation regularization to prevent student features from being redundant. Specifically, CD exploits the decorrelation mechanism (the second term of Eq.2) to prevent information collapse in the student model. However, IC loss does not utilize such an information maximization mechanism, and only relies on the global one-to-one MSE distance (L2-norm) between modality-specific self-correlation matrices. Hence, CD pursues a different learning principle compared to TiG-BEV.

These differences lead to the competence of CD over TiG-BEV. Tab.9 shows that CD effectively reduces duplicate dimensional feature components compared to the correlation-based distillation of TiG-BEV. Furthermore, in Tab. 8, CD ultimately achieves performance improvements over TiG-BEV and BEVDistill by up to +3.2% of NDS and +3.9% of mAP. These results consistently demonstrate the significance of mitigating feature redundancy in cross-modal knowledge distillation and enhancing feature quality through our feature-level distillation approaches.

Table 9: Validation for correlation regularization effect of CD loss. For both TiG-BEV and our STXD, we measured the effective dimension ($d_{\text{eff}}$) and dimensional redundancy values defined in Sec.4.2. TiG-BEV (FD) represents the results of applying only the feature distillation loss from TiG-BEV.

| Method | $d_{\text{eff}}\uparrow$ | $d_{\text{eff}}^2\uparrow$ | Dim.Red.$\downarrow$ |
|---|---|---|---|
| TiG-BEV (FD) [22] | 1.620 | 2.634 | 0.174 |
| **STXD (CD)** | **1.694** | **2.871** | **0.062** |
| Teacher [33] | 2.194 | 4.813 | 0.038 |

## B.4 Qualitative Results

### B.4.1 Prediction Results on BEV Space

Fig.5 and 6 provide examples of 3DOD results in the BEV space from the LiDAR-based model (top-left), the multi-view camera-based model trained with our STXD (top-middle), and the camera-based model trained without knowledge distillation (camera-only model, top-right). Additionally, at the bottom of each example, we have included camera views that correspond to the highlighted regions, which can help in better understanding the results. Overall, STXD enables more accurate prediction of bounding box locations, scales, and orientations compared to the camera-only model (highlighted by red dashed boxes). This observation is consistent with the quantitative results in Tab.4 and 6 of the manuscript. STXD also leads to a reduction of false positives. In certain examples, we observe a decrease in false positives even when comparing the results to the LiDAR-based model (highlighted by pink dashed boxes). Notably, in Fig.6, STXD effectively reduces false positive detections in 3DOD compared to the LiDAR-based model and the camera-only model. These results indicate that our STXD framework effectively transfers valuable information from the LiDAR-based teacher to the camera student, leading to improved detection performance.

### B.4.2 Qualitative Comparison on BEV Features

As an extension of Fig.1 in the manuscript, we provide additional examples of BEV features learned using different feature-level distillation methods in Fig.7. In various samples, the model trained with the correlation regularizing distillation (CD) consistently generates more similar feature patterns (Fig.7-(b)) to the teacher model (Fig.7-(a)) compared to other methods (Fig.7-(c) and (d)). These results demonstrate the effectiveness of the proposed method in transferring the rich information contained in the teacher model to the student model.

### B.4.3 Temporal Similarity Maps

As an extension of Fig.4 in the manuscript, Fig.8 illustrates the learning progress of the temporal similarity map during the training phase for different values of $k$ ($k = 1, 2, 3$). As the training progresses, the cross-modal temporal similarity map $\mathbf{S}^{(-k)}$ becomes similar to the intra-modal temporal similarity $\mathbf{T}^{(-k)}$. This indicates that the student model effectively learns the temporal knowledge encoded in the similarity maps from the teacher, and ultimately leads to significant improvements in 3DOD, as demonstrated in our extensive experiments reported in the manuscript.

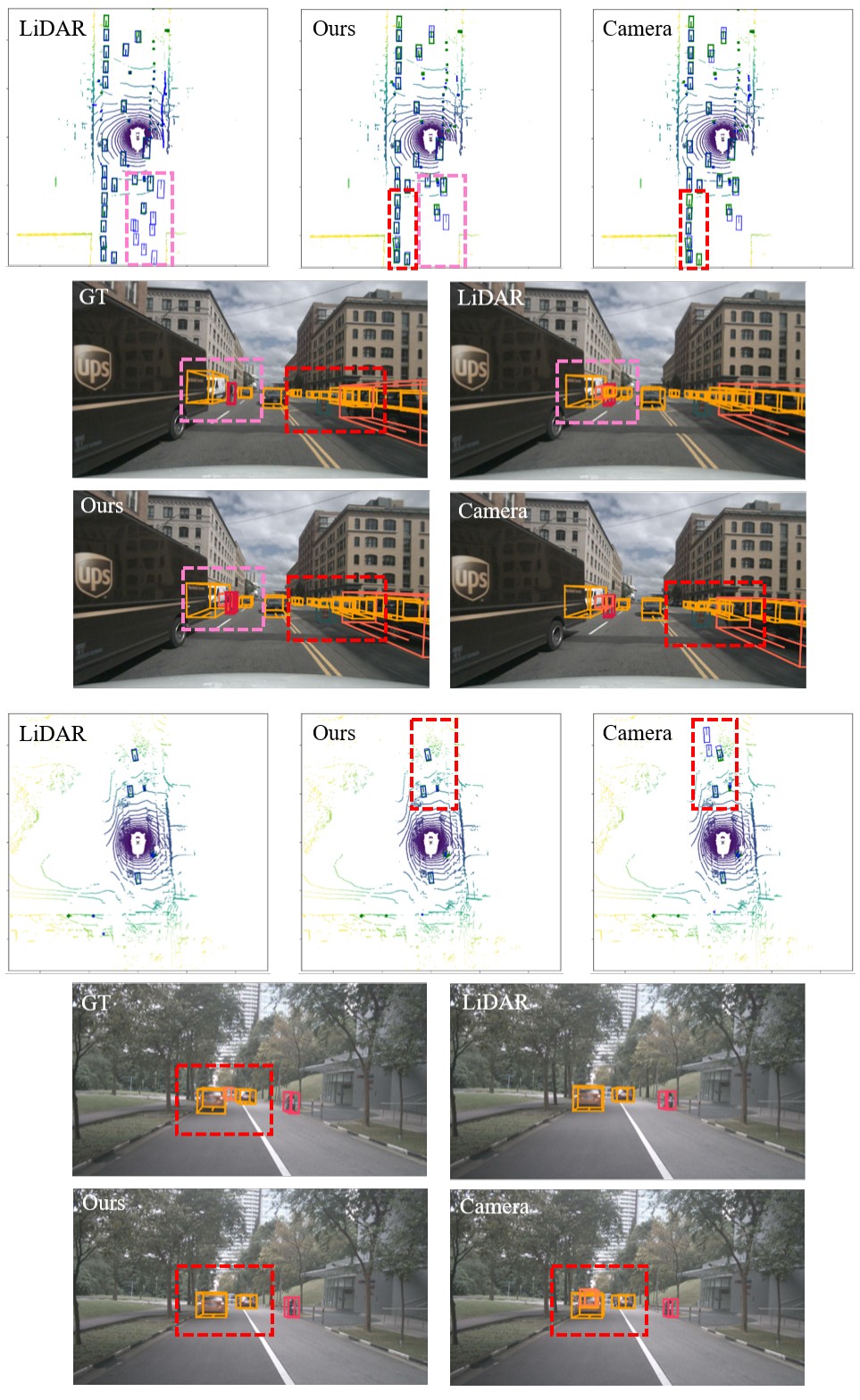

Figure 5: Examples of 3D object detection results from LiDAR-based model (top-left), multi-view camera-based model with STXD (top-middle), and camera-only model (top-right). Green rectangles indicate ground truth bounding boxes, while blue rectangles represent the predicted bounding boxes. Pink dashed boxes highlight the comparisons of STXD with the LiDAR-based model, while red dashed boxes highlight the comparisons with the camera-only model. We also provide camera views corresponding to the highlighted regions at the bottom.

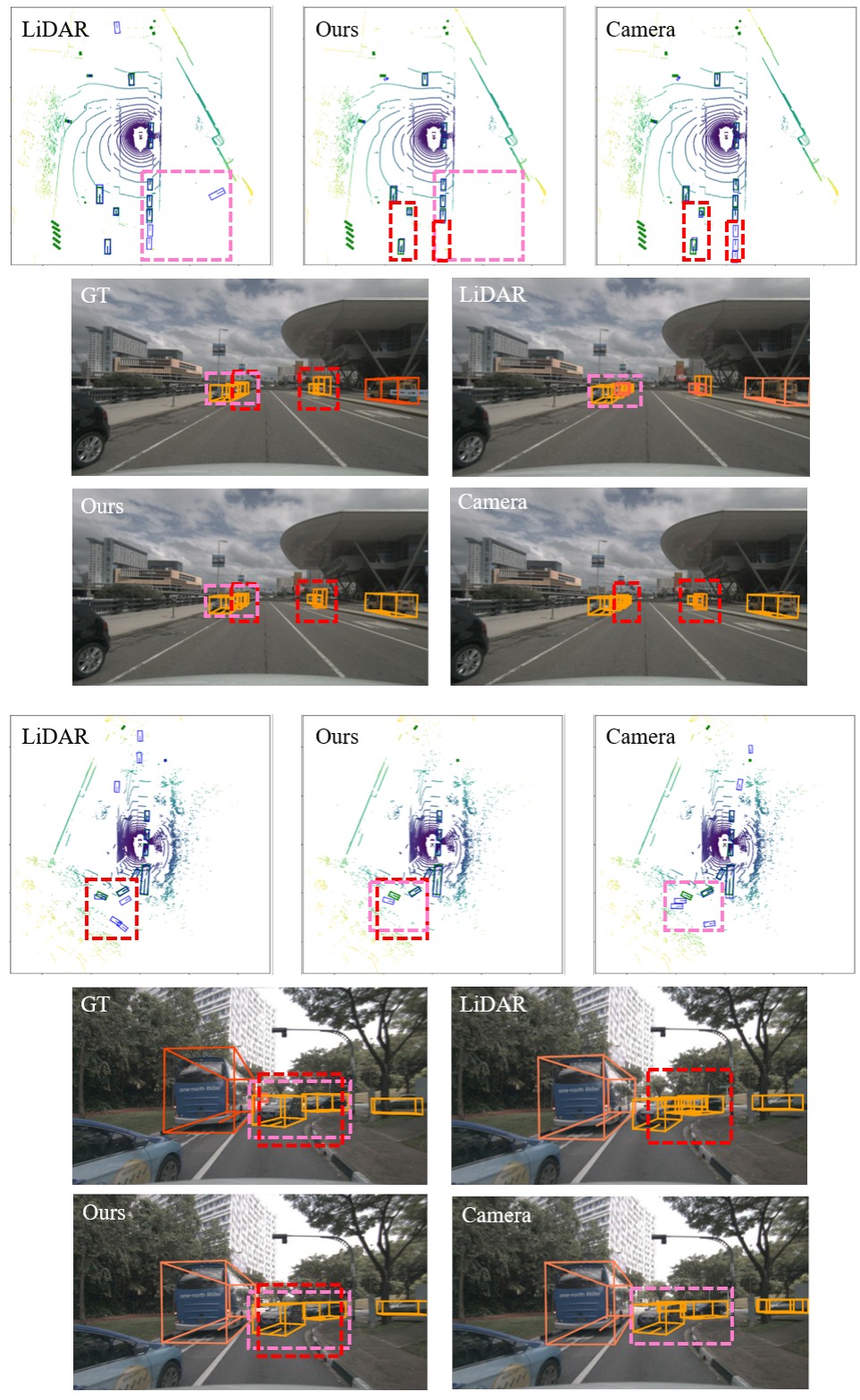

Figure 6: Examples of 3D object detection results from LiDAR-based model (top-left), multi-view camera-based model with STXD (top-middle), and camera-only model (top-right). Green rectangles indicate ground truth bounding boxes, while blue rectangles represent the predicted bounding boxes. Pink dashed boxes highlight the comparisons of STXD with the LiDAR-based model, while red dashed boxes highlight the comparisons with the camera-only model. We also provide camera views corresponding to the highlighted regions at the bottom.

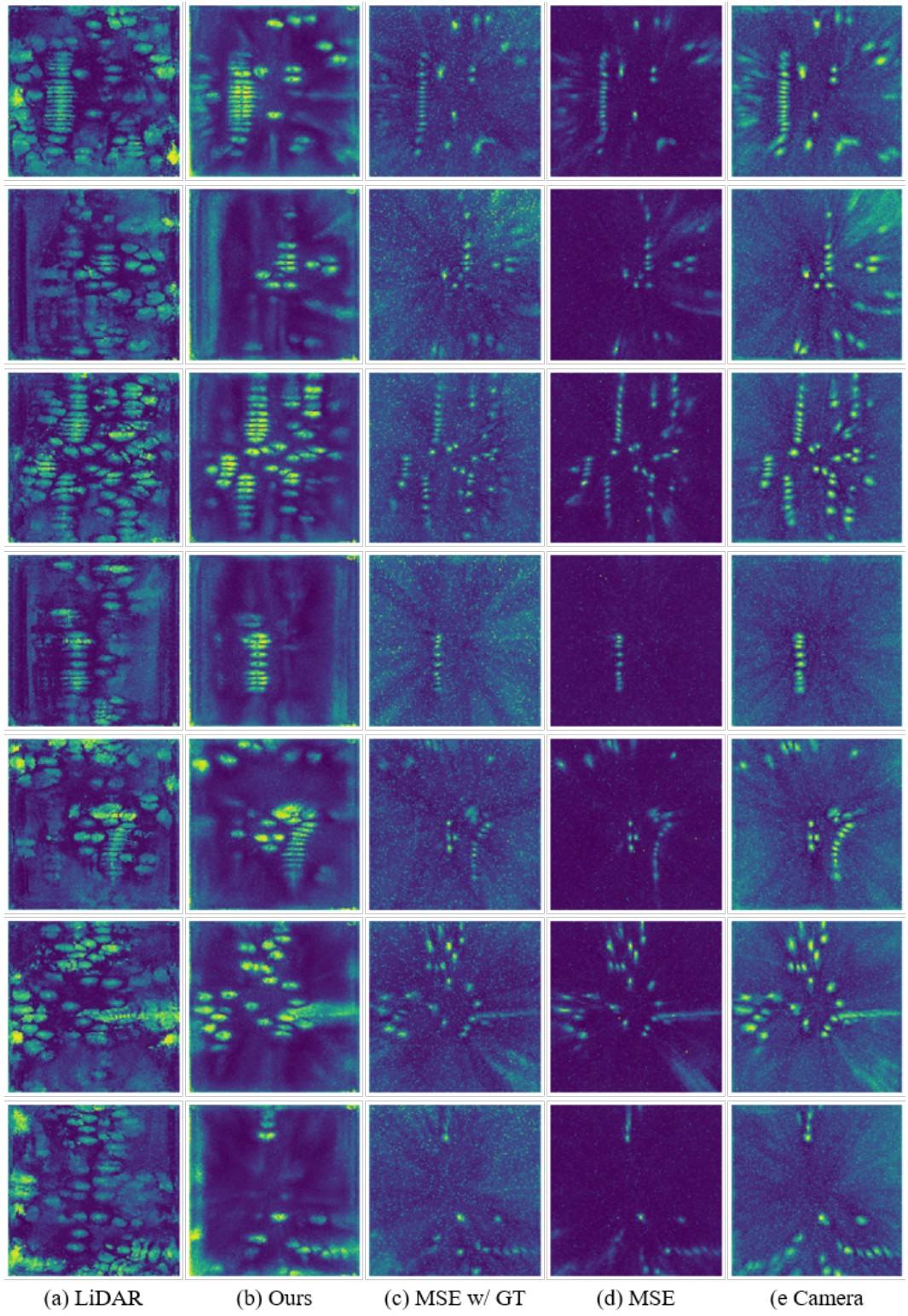

(a) LiDAR  (b) Ours  (c) MSE w/ GT  (d) MSE  (e Camera

Figure 7: Examples of BEV features generated by **(a)** the LiDAR-based model, **(b)-(d)** the multi-view camera-based model trained with different feature-level distillation methods, and **(e)** the camera-only model. We visualize the $\mathcal{L}_2$-norm value of BEV features.

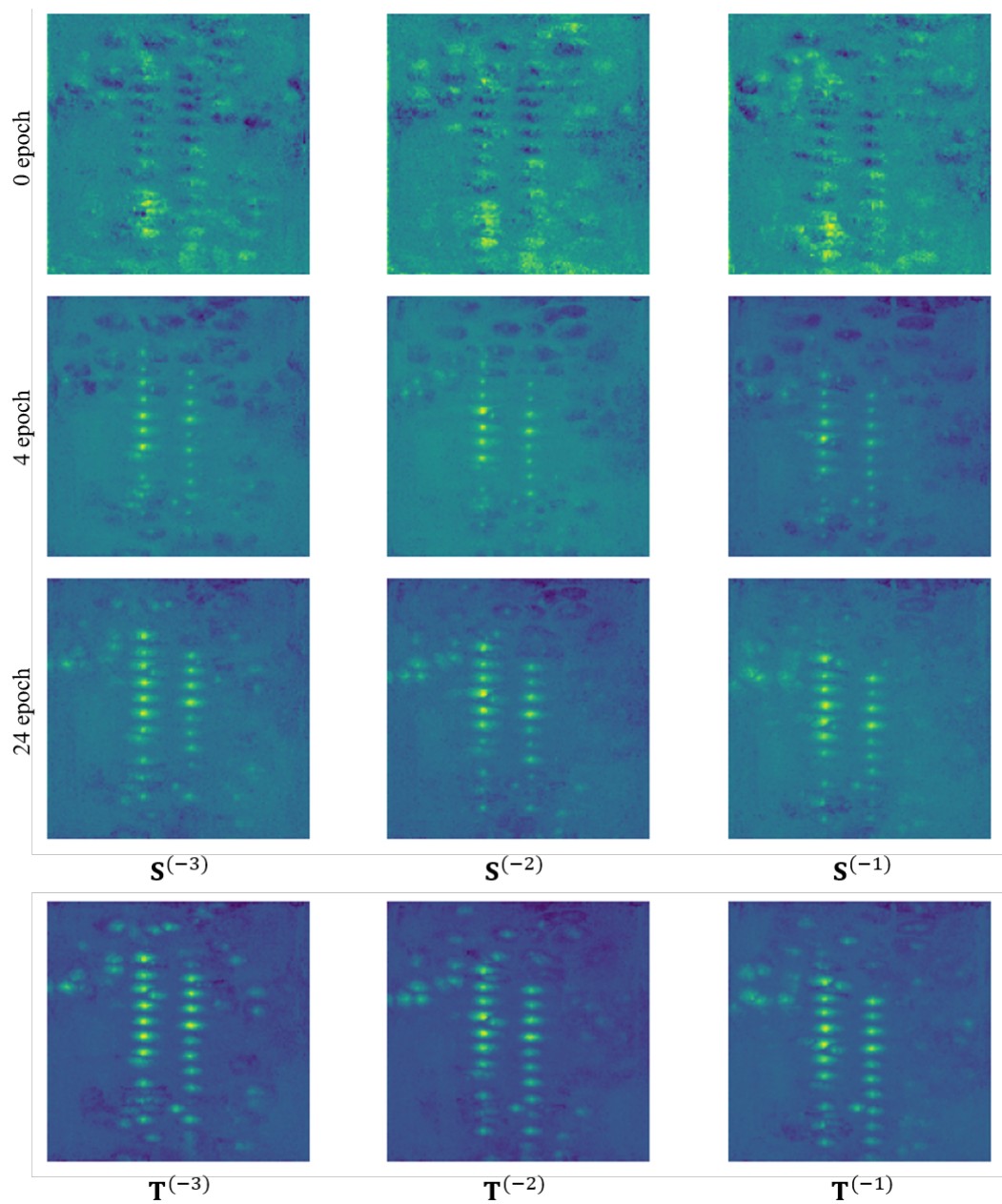

Figure 8: Examples of the learning progress for the temporal similarities during the training phase.

