# OpenReview forum: "STXD: Structural and Temporal Cross-Modal Distillation for Multi-View 3D Object Detection"
_NeurIPS.cc/2023/Conference — NeurIPS 2023 poster_

### Official Review · Reviewer_U3eQ · 2023-06-27

**Soundness:** 3 good
**Presentation:** 3 good
**Contribution:** 3 good
**Rating:** 5
**Confidence:** 5

**Summary:**

This paper presents a novel cross-modal knowledge distillation framework for multi-view 3D object detection, namely STXD, with three effective strategies.

(1) correlation-based cross-modal feature distillation, which gets rid of the domain gap on the feature level during distillation.

(2) feature similarity guided temporal feature distillation, which considers the dynamic scenes (especially moving objects) across multiple frames through similarity maps.

(3) response-level distillation via bipartite matching between teacher and student predictions.

Overall, STXD significantly improves current 3D detectors with its effective KD framework on nuScenes dataset, and provide a clear architecture.

**Strengths:**

1. The paper is well-written and easy to follow.

2. The paper presents new perspectives on knowledge distillation for multi-view 3D detectors compared to recent works, i.e., temporal-based feature distillation.

3. The improvements on nuScenes dataset validate the effectiveness of STXD and the experiments on nuScenes dataset are extensive.

4. The paper is a practical work with solid enhancement to current 3D detectors.

**Weaknesses:**

1. The main concern of the paper is the novelty of this paper. Actually, the correlation-based feature distillation to decouple the domain knowledge for LiDAR and image is not novel, even in the cross-modal distillation (for example TiG-BEV[1], on arxiv 5 months ago). The response-level distillation is also similar to BEVDistill [2].

[1] TiG-BEV: Multi-view BEV 3D Object Detection via Target Inner-Geometry Learning
[2] BEVDistill: Cross-Modal BEV Distillation for Multi-View 3D Object Detection

**Questions:**

1. The paper presents remarkable improvements on BEVFormer and UVTR, however, there are other types of detectors, for instance, BEVDepth4D, BEVDet4D-Depth. What about these LSS-based detectors? Can STXD improve them even in a stronger setting? (e.g. with a ConvNeXt-based backbone and achieves 60+NDS on nuscenes test set?)

2. What about the performance on Waymo dataset? The authors are suggested to conduct experiments on other datasets except nuScenes to make up for the limited novelty in this paper.

---

> ### Author Rebuttal · Authors · 2023-08-09
>
> We appreciate the recognition from reviewer "U3eQ" for acknowledging our "new perspectives on knowledge distillation for 3D detectors," particularly our "temporal-based feature distillation" approach. "U3eQ" also noted the effectiveness of STXD, validated through extensive experiments, and its "practicability" and "solid/remarkable (performance) enhancement". Below, we address your concerns regarding the contributions of STXD, including Correlation Regularizing Distillation (CD) and Response-Level Distillation (RD).
>
> **Correlation Regularizing Distillation (CD) vs. TiG-BEV [R3]:**\
> Our Correlation Regularizing Distillation (CD) and TiG-BEV [R3] both consider dimensional correlation of features, which might appear to be similar at first glance. However, we want to emphasize that these methods are distinct in terms of their underlying objectives and implementations. Consequently, there exist clear differences between CD and TiG-BEV in several aspects. TiG-BEV proposes two correlation-based losses:
>
> $L_{\text{inter-channel}}=\|\|C_F - C_G\|\|_2$, where $C_F=F F^T, C_G=G G^T \in \mathbb{R}^{D\times D}$,
>
> $L_{\text{inter-keypoint}}=||\tilde{C}_F - \tilde{C}_G||_2$, where $\tilde{C}_F=F^T F, \tilde{C}_G=G^T G \in \mathbb{R}^{N\times N}$.
>
> Both losses try to minimize distance between **intra-modal self-correlation** of BEV features from the teacher ($F$) and student models ($G$) based on the MSE distance function.
>
> In contrast, the core contribution and novelty of CD lies in its introduction of **cross-modal cross-correlation regularization** to maximize the information transferred from the teacher to the student. Specifically, our inspiration comes from recent self-supervised learning methods [R1, R2], which utilize a **decorrelation mechanism** to reduce duplicated information in the learned features. We found such info-max approach interesting and valuable, especially for knowledge distillation of the BEV features containing noisy and redundant background information. Our CD loss is defined as follows (quoted from Eq.2):
>
> $L_{\text{CD}}= \sum_{i} (1-C(i,i))^2 + \lambda_{c} \sum_{i} \sum_{j\neq i} C(i,j)^2, \text{where } C = \hat{F}^T \hat{G} \in \mathbb{R}^{D \times D}.$
>
> CD loss utilizes the decorrelation mechanism (the second term of Eq.2) to prevent information collapse in the student model, while enhancing the dimensional component-wise feature similarities (the first term of Eq.2) (refer to L43-L48 and L141-L145). However, TiG-BEV does not utilize such an information maximization mechanism. For a clear comparison, we have also depicted relevant diagrams to clarify these differences in Fig.R2 of the rebuttal PDF. These differences lead to the competence of CD over TiG-BEV, as presented in Tab.R1 and Tab.R2. From these results, we confirmed that (1) CD effectively reduces duplicate dimensional feature components compared to the correlation-based distillation of TiG-BEV (Tab.R2), and (2) ultimately outperforms prior cross-modal distillation methods (i.e., improvements over TiG-BEV [R3] and BEVDistill [R6] by up to +3.2% of NDS and +3.9% of mAP (Tab.R1)). Such results are consistent with our extensive component-wise ablation and main results presented throughout the manuscript and the Appendix.
>
> **Additional Contribution of Response-Level Distillation (RD):**\
> We are also aware of that prior works [R6-R9] have introduced similar approaches to RD. Please note that we acknowledge that and do not claim the introduction of RD as our main contribution. Rather we explicitly indicated our "adoption" and "inspiration" from such output distillation strategies to further enhance our knowledge distillation performance (L59-L61, L167-L168). Please note the improvement gained from our core feature-level distillation methods (CD,TD) is larger than RD (i.e., gain of NDS(%) and mAP(%) over based student model BEVFormer [R4]: RD=1.44% and 1.56%, TD=1.77% and 2.15%, and CD=2.47% and 2.75%) as shown in Tab.1. More importantly, STXD without RD already outperforms BEVDistill [R6] (i.e., STXD w/o RD=54.0% of NDS and 43.4% of mAP in Tab.1 vs. BEVDistill=52.4% of NDS and 41.7% of mAP in Tab.4). Hence, we can consider RD as an add-on method for improved cross-modal knowledge distillation, given its widespread use in most state-of-the-art distillation methods for object detection [R6-R9].
>
> To draw a clear comparison with BEVDistill, our output distillation formulation is primarily inspired by the quality-score approaches used in output-guided knowledge distillation for 2D object detectors [R7,R9] (refer to L171-L177). In contrast, BEVDistill employs a critic network to approximate distributional similarities in output-level features between teacher and student models. Hence, technically speaking, RD is different from BEVDistill.
>
> **Additional Baseline for Student Model:**\
> Please note that we extensively validated our STXD framework with two of baseline student models, BEVFormer [R4] and UVTR-C/CS [R5], representing attention-based and LSS-based BEV feature generation approaches, respectively. To further validate the effectiveness of our approach, we also present additional validation experiments with BEVDepth [R11] in Tab.R1 of the rebuttal PDF, where CD and TD enhances BEVDepth, by achieving notable improvements of +5.3% and +5.2% in terms of NDS, respectively.
>
> **Additional Validation on Waymo Dataset:**\
> Generally, the method that performs well on the nuScenes dataset often achieves favorable outcomes in the Waymo dataset, as observed in [R4,R15]. Therefore, we believe our extensive validation and testing results on the nuScenes sufficiently demonstrate the effectiveness of our cross-modal distillation approach. To accommodate your suggestion, we are conducting experiments on the Waymo dataset, but we are unable to report the results from such a large-scale dataset (230K frames over 1TB) within the short rebuttal period. We will report results in the final version of our paper.

---

> ### Comment · Reviewer_U3eQ · 2023-08-19
> **Response to authors**
>
> Thanks for the responses provided by the authors. After reading the rebuttal, the reviewer decided to slightly raise its score to borderline accept. However, the author is strongly suggested to provide experiments on Waymo datasets in the future version to validate the effectiveness of their approaches. :)

---

### Official Review · Reviewer_X2eU · 2023-07-04

**Soundness:** 3 good
**Presentation:** 3 good
**Contribution:** 3 good
**Rating:** 5
**Confidence:** 4

**Summary:**

This work presents a cross-modal knowledge distillation framework for the multi-view, called STXD. It tries to distill structural and temporal knowledge across different modalities (from LiDAR to Camera) from three aspects, namely Correlation Regularizing Distillation (CD), Temporal Consistency Distillation (TD), and Response-Level Distillation (RD). Experiments prove the effectiveness of the proposed approach.

**Strengths:**

1. This work focuses on distilling structural features from LiDAR to the camera-based setting, which is promising and could have potential usage for application.
2. The method to utilize temporal information via feature comparison is interesting.
3. Experiments are sufficient and show significant improvement over the baseline on several benchmarks.
4, Overall, the writing is clear and easy to follow.

**Weaknesses:**

1. It's interesting to apply correlation regularization to cross-modality distillation. However, the experimental comparison with classic feature-level distillation and analysis are missing. I'm wondering why the correlation regularization achieves much better results than the classic one.
2. Because temporal info seems to bring better results. How about the performance with a longer temporal sequence and stronger benchmark, like SOLOFusion [37].
3. Some typos should be fixed: L58 This way-> In this way.

**Questions:**

Please refer to the weakness section.

**Limitations:**

The authors discuss the social impact in the paper.

---

> ### Author Rebuttal · Authors · 2023-08-09
>
> We thank the reviewer "X2eU" for acknowledging our cross-modal distillation framework is "promising" and has "potential usage for application" as well as recognizing our "sufficient experiments" showing "significant improvements" over the baselines. We hope you find our responses provided below helpful to address your questions.
>
> **Comparison with "classic feature-level distillation" and Analyses:**\
> To our best understanding, we interpret "classic feature-level distillation" referred to by "X2eU" is closely related to conventional MSE-like global one-to-one distance minimization methods [R6-R9]. If so, please note that we provided the comparisons between our Correlation Regularizing Distillation (CD) and conventional feature-level distillation approaches in Tab.2. In particular, we showed that CD outperforms the conventional "MSE" and "MSE w/ GT" distillation methods by up to +2.24% of NDS and +2.25% of mAP.\
> We discussed that the observed performance gain over conventional feature-level distillation methods can be attributed to maximizing the information in the student features learned from the teacher (L142-L145). To validate our argument, we presented two quantitative measures: (1) Effective Dimension and (2) Dimension Redundancy (L254-L257). The results from Tab.3 and Fig.3 depicted the reduction of redundant feature components as we anticipated. In addition, Fig.1 and Fig.A3 of Appendix demonstrated the improved quality of feature similarities between the teacher and the student compared to classical distillation approaches (e.g., MSE and MSE w/ GT sampling).
>
> **Further validation with a longer temporal sequences:**\
> Thanks for suggesting such an interesting and valuable extension of our temporal distillation approach. We plan to apply our Temporal Consistency Distillation (TD) over longer temporal sequences and this could lead to another interesting contribution to the field of temporal cross-modal distillation and 3D object detection.
>
> Lastly, we will correct typos in the final version.

---

> > ### Comment · Reviewer_X2eU · 2023-08-21
> > **Response to authors**
> >
> > Thanks for the rebuttal provided by the authors. It addresses most of my concerns. So, I'd like to keep my original rating as Borderline accept.

---

### Official Review · Reviewer_XsGg · 2023-07-04

**Soundness:** 2 fair
**Presentation:** 4 excellent
**Contribution:** 3 good
**Rating:** 5
**Confidence:** 5

**Summary:**

This paper proposes a novel structural and temporal cross-modal knowledge distillation (STXD) framework for multi-view 3D Object Detection. STXD leverages the Correlation Regularizing Distillation loss to address the distributional divergences between non-homogeneous modalities. Besides, it effectively distills temporal information contained in previous frames of the teacher and utilizes response-level distillation to further improve the quality of knowledge distillation at the prediction-level. Extensive experiments on nuScenes show that STXD significantly improves the NDS and mAP of the base student detectors.

**Strengths:**

1. The paper provides valuable insights into addressing distributional divergences between non-homogeneous modalities through the use of the Correlation Regularizing Distillation loss.
2. The effectiveness of the Correlation Regularizing Distillation loss is well demonstrated through ablation studies.
3. The writing is clear and the paper is easy to follow.

**Weaknesses:**

1. The novelty of Response-Level Distillation (Sec 3.4) is limited, as a very similar method is proposed in BEVDistill [1].
2. The effectiveness of Temporal Consistency Distillation is somewhat uncertain. Many previous works demonstrate the effectiveness of using multi-frames [2-4], and their methods are relatively simple (such as concatenating the temporal BEV features) yet effective in achieving state-of-the-art performance. However, this paper does not include experiments to demonstrate the effectiveness of Temporal Consistency Distillation in models that utilize multi-frames. If the authors can provide these experiments, I may raise my rating.
3. While the insight of using the Correlation Regularizing Distillation loss to address distributional divergences is valuable, it should be noted that the authors do not propose the Correlation Regularizing Distillation loss itself.

[1] Zehui Chen, Zhenyu Li, Shiquan Zhang, Liangji Fang, Qinhong Jiang, and Feng Zhao. BEVDistill: Cross- modal BEV distillation for multi-view 3d object detection. ICLR 2023.
[2] Chunrui Han, Jianjian Sun, Zheng Ge, Jinrong Yang, Runpei Dong, Hongyu Zhou, Weixin Mao, Yuang Peng, Xiangyu Zhang. Exploring Recurrent Long-term Temporal Fusion for Multi-view 3D Perception.
[3] Jinhyung Park, Chenfeng Xu, Shijia Yang, Kurt Keutzer, Kris Kitani, Masayoshi Tomizuka, Wei Zhan. Time Will Tell: New Outlooks and A Baseline for Temporal Multi-View 3D Object Detection.
[4] Junjie Huang, Guan Huang. BEVDet4D: Exploit Temporal Cues in Multi-camera 3D Object Detection.

**Questions:**

1. The paper does not clearly explain how to utilize sweep images in the student models. Additionally, it's worth noting that most of the camera-based baseline models seem not to use multi-frames, as evidenced by the relatively high mATE (>0.6) and mAVE (>0.5) values.
2. Could the authors clarify why CBGS is not employed, given its potential to facilitate full model training?
3. Could the authors offer the results on the nuScenes test sets? (Leaderboard web: https://www.nuscenes.org/object-detection?externalData=all&mapData=all&modalities=Camera)

**Limitations:**

There are still many bad cases when using camera based 3D object detection. Achieving high performance with most of the existing methods often requires significant computational resources, such as training on 8 NVIDIA A100 GPUs.

---

> ### Author Rebuttal · Authors · 2023-08-09
>
> We appreciate the acknowledgment from reviewer "XsGg" for recognizing "effectiveness" and "valuable insights" of our non-homogeneous cross-modal knowledge distillation framework, particularly the use of Correlation Regularizing Distillation (CD) as well as our comprehensive ablation studies. Below, we provide answers to your questions to further address your concerns.
>
> **Response-Level Distillation (RD):**\
> As pointed out by "XsGg," we are also aware of that similar approaches to our response-level distillation have been introduced in prior works [R6-R9]. **However, please note that we acknowledge that and do not claim the introduction of RD as our main contribution**. Rather, we explicitly indicated our "adoption" and "inspiration" from such output distillation strategies to further enhance our knowledge distillation performance (L59-L61, L167-L168). Please also note that **the improvement gained from our core feature-level distillation methods (CD,TD) is larger than RD** (i.e., gain of NDS(%) and mAP(%) over based student model BEVFormer [R4]: RD=1.44% and 1.56%, TD=1.77% and 2.15%, and CD=2.47% and 2.75%). Moreover, **STXD without RD already outperforms BEVDistill [R6]** (STXD w/o RD=54.0% of NDS and 43.4% of mAP in Tab.1 vs. BEVDistill=52.4% of NDS and 41.7% of mAP in Tab.4). Hence, we can consider RD as an add-on method for improved cross-modal knowledge distillation, given its widespread use in most state-of-the-art distillation methods for object detection [R6-R9].
>
> To draw a clear comparison with BEVDistill, our output distillation formulation is primarily inspired by the quality-score approaches used in output-guided knowledge distillation for 2D object detectors [R7,R9] (L171-L177). In contrast, BEVDistill employs a critic network to approximate distributional similarities in output-level features between teacher and student models. Hence, technically speaking, RD is different from BEVDistill.
>
> **Temporal Consistency Distillation on Multi-Frame Models:**\
> In our experiments, we adopted two baseline student models that utilize multi-frame inputs. In UVTR-CS [R5], multi-frame images are concatenated and forwarded to the image encoder. BEVFormer [R4] generates the BEV feature of the current frame by referring to the BEV feature of the past frames via temporal self-attention mechanism. We followed the implementation of each student models and incorporated the past three frames in cross-modal distillation learning. As shown in Tab.1, 4, 5, and 6, TD demonstrated its effectiveness when it is incorporated with student models that utilize multi-frame  (i.e., BEVFormer and UVTR-CS) as well as single-frame (i.e., UVTR-C). In the final version, we will provide further clarification regarding the implementation details of the student models in Sec.4.1.
>
> **Correlation Regularizing Distillation (CD):**\
> We indeed did not create the decorrelation mechanism itself [R1,R2] in current submission. However, it also should be noted that we are the first to bring such self-supervised representation learning approach into the field of cross-modal distillation and 3D object detection based on a strong motivation (L36-L48). As “XsGg” also agreed with, our component-wise validation provides informative and valuable insights in this direction. We hope the follow-up research considers to use our CD loss instead of MSE-like loss functions based on the new perspectives and proof of effectiveness of CD presented in our extensive experiments.
>
> **Further Clarifications:**
> - Above, we have explained how our baseline student models (BEVFormer [R4], UVTR-CS [R5]) utilize multi-frame inputs. Additionally, for mATE and mAVE metrics, applying STXD (especially for TD in Tab.1 and Tab.5) results in an overall enhancement compared to the baselines and other KD methods (Tab.4 and 6). It is noteworthy that most reviewers, "4Udp", "X2eU", and "U3eQ", also noted the effectiveness and novelty of our temporal cross-modal distillation approach.
> - Regarding CBGS [R10], which mainly focuses on a sampling strategy for training, it has limited relevance to knowledge distillation. To isolate and specifically validate the effectiveness of KD methods without any potential compounding effects, we did not utilize CBGS during training (L326). Furthermore, applying CBGS to our baseline model, BEVFormer, led to an extension of 1 epoch to 4.5 epochs, resulting in the entire training time exceeding two weeks even on NVIDIA A100 GPUs. Therefore, we find it impractical and challenging to apply CBGS to our baseline student models, making it difficult to ensure reproducibility.
> - Tab.6 presents the evaluation results on the nuScenes test set, which can also be verified on the official eval.ai webpage (https://eval.ai/web/challenges/challenge-page/356/overview). We included captured results from the webpage in Fig.R1 of the rebuttal PDF for reference. Reviewers can also find our results by searching the keyword "STXD" in the official leaderboard (https://eval.ai/web/challenges/challenge-page/356/leaderboard/1012). Upon acceptance, we will request the authorities to display our testing results on the nuScenes dataset webpage, as you have suggested.

---

> > ### Comment · Reviewer_XsGg · 2023-08-20
> > **Additional questions**
> >
> > Thanks the authors for the feedback. I still have some questions:
> > 1. The experiment on BEVDepth is not sufficient. While many methods demonstrate better performance than BEVDepth with fewer parameters and lower image resolutions, they may not perform well when the settings become more complex. It is more convincing that the authors make comparison with BEVDepth on the leaderboard.
> > 2. This method does not well validate its effectiveness in integrating long-term temporal information, which is essential for accurate 3D detection.

---

> > > ### Author Response · Authors · 2023-08-20
> > >
> > > We thank the reviewer “XsGg” for dedicating the time to review our rebuttal. We are pleased to provide additional clarifications below to address your questions.
> > >
> > > **Additional validation on BEVDepth**\
> > > The authors also acknowledge the potential benefits of including BEVDepth [R11] results using relatively heavier and more complex settings (i.e., heavier backbones and larger image resolutions). Nonetheless, for a swift and fair comparison with previous methods, including TiG-BEV [R3] and BEVDistill [R6], we conducted additional experiments with BEVDepth using the same setup as TiG-BEV (i.e., ResNet-50 as the image backbone in Tab.R1 of the rebuttal PDF). As the evidence of the effectiveness of our framework under heavier and complex settings, we have achieved significantly improved performance across both larger (i.e., V2-99, ResNet-101-DCN from Tab.6) and smaller backbone variants (i.e., ResNet-18 and 50 from Tab.A1 in the Appendix) for the BEVFormer and UVTR-C/CS baselines. Moreover, we reported the results from the heavier setting on the nuScenes leaderboard. We hope such validations involving variant settings appear to be reasonable to the reviewer “XsGg” to ensure the effectiveness of our framework in various settings.
> > >
> > > **Validation with long-term temporal information**\
> > > While a specific count of past frames defining 'long-term temporal information' is not provided, it is noteworthy that BEVDepth leverages relatively extended temporal information by concatenating an aligned BEV feature randomly selected from the past 3 to 9 frames (employing the temporal strategy of BEVDet4D [R16]). Through our additional experiments with BEVDepth in Tab.R1 of the rebuttal PDF, we also validated the effectiveness of both CD and TD with relatively longer frames of inputs, in contrast to BEVFormer [R4] and UVTR-CS [R5] exploiting only the past 3 frames. To the best of our knowledge, we are the first to offer such new perspectives on structural and temporal cross-modal distillation for 3D object detection. In future work, we plan to further apply and validate our approach to other models utilizing long-term information such as [R17] and [R18] as suggested by ”XsGg”. We will discuss this plan in Conclusion section of the final version.
> > >
> > > [R16] Huang, J., & Huang, G. (2022). BEVDet4D: Exploit temporal cues in multi-camera 3d object detection. arXiv preprint arXiv:2203.17054.\
> > > [R17] Han, C., Sun, J., Ge, Z., Yang, J., Dong, R., Zhou, H., ... & Zhang, X. (2023). Exploring recurrent long-term temporal fusion for multi-view 3d perception. arXiv preprint arXiv:2303.05970.\
> > > [R18] Park, J., Xu, C., Yang, S., Keutzer, K., Kitani, K. M., Tomizuka, M., & Zhan, W. (2023). Time Will Tell: New Outlooks and A Baseline for Temporal Multi-View 3D Object Detection. In The Eleventh International Conference on Learning Representations.

---

> > > > ### Comment · Reviewer_XsGg · 2023-08-22
> > > >
> > > > I think that the Correlation Regularizing Distillation loss which tackles distributional divergences between non-homogeneous modalities in 3D detection deserves broader recognition, and I thus slightly raise my score to borderline accept. However, I still believe that it is important to conduct extended experiments involving long-term temporal information (more than 8 frames) and explore complex settings related to BEVDepth. I highly recommend the authors to incorporate these results in the next version.

---

> > > > > ### Author Response · Authors · 2023-08-22
> > > > >
> > > > > We appreciate the efforts of reviewer "XsGg" in reviewing our responses. We are delighted that the novelty of the proposed CD loss has been acknowledged. We will definitely attempt to incorporate the experiments suggested by "XsGg" in the next version.

---

### Official Review · Reviewer_1M3u · 2023-07-06

**Soundness:** 2 fair
**Presentation:** 3 good
**Contribution:** 1 poor
**Rating:** 4
**Confidence:** 5

**Summary:**

This paper introduces a cross-modal distillation framework named STXD, which addresses existing distillation problems from both structural and temporal perspectives. Structure-based distillation imposes constraints on the similarity matrix of the student and teacher, ensuring that corresponding positions have a higher similarity. Temporal distillation is accomplished by distilling the feature similarity matrix over time. Ultimately, response-level distillation is employed to further improve the performance. Experiments on the nuScenes dataset demonstrate that the proposed approach outperforms previous distillation techniques.

**Strengths:**

+ The proposed cross-modal distillation from LiDAR to image works well (a marked improvement ranging from 2.8% to 4.5%).
+ Experiments illustrate that the proposed approach outperforms previous approaches on the nuScenes dataset.
+The paper is well written with a clear description of the method and comprehensive component analysis.

**Weaknesses:**

- The concept of 'structural distillation' seems no distinction from existent Mean Squared Error (MSE) distillation. It applies a weight of 0.01 to different positioning areas, hence, the leading role is still played by the loss at corresponding positions, which essentially substitutes MSE with a measure of similarity.
- The structural distillation appears not reasonable. In fact, spatially adjacent tokens should have some correlation. In Fig.2, enforcing the correlation of corresponding positions to 1, and other positions to 0 is unjustified. Additionally, correlation is unable to respond to the redundancy of features, so Fig.3 is void of significance.
- The paper employs an abundance of similarity matrices, resulting in substantial consumption of memory during the training process. I have noticed that BEVFormer uses only a 128x128 Bird's Eye View (BEV) resolution in the experiments. Whether the computational constraints caused this deviation from the default settings of BEVFormer.
- The motivation for temporal distillation is insufficient. Theoretically, both the teacher and student features of the current frame contain temporal information. Thus, it brings into question the need to formulate the distillation approach in this manifestation.
- It is better to conduct distillation based on stronger baselines such as BEVDepth, following BEVDistill. Table 2 ought to incorporate some comparison with prevalent distillation methods in 2D detection, such as GID. Techniques such as CLIP are seldom used. For the generality of distillation, the distillation on the Waymo dataset is indispensable.

**Questions:**

1. The notion of 'inherent structural knowledge' has not been succinctly explicated. For example, whether it refers to the 'instance'. It appears no relation with the term 'structure'.
2. As described in the weakness, the authors should give a reasonable explanation for the motivation of structural distillation and temporal distillation.
3. More experiments are needed to prove the effectiveness of the method.

**Limitations:**

The authors have already provided the limitations in the paper.

---

> ### Author Rebuttal · Authors · 2023-08-09
>
> We thank the reviewer "1M3u" for taking time and effort to review our work. We also appreciate the reviewer's recognition of the competence of our knowledge distillation approach, clear presentation, and comprehensive component-wise analyses. Below, we provide responses to your concerns, and we hope they are considered reasonable and acceptable.
>
> **Structural Distillation through Correlation Regularization (CD):**\
> First and foremost, we clarify the motivation and concept underlying our Structural Distillation method. As shown in Fig.1-(a) and (e), different modalities exhibit distinctive feature distributions. However, MSE-like simple cross-modal distillation losses (e.g., L2-Norm) primarily focus on position-wise spatial correspondence and do not take into account dimension-wise feature structures (e.g., the distribution of dimensional feature space). Consequently, such MSE-like losses fail to adequately transfer the structure of the feature space (i.e., the “structural knowledge” inherent in modality-specific feature space), as shown in Fig.1-(c) and (d). To address these challenges, we propose a new structural distillation method that transfers the knowledge of dimension-wise feature structures (L36-L48).
>
> To our best understanding, however, “1M3u” construed our structural distillation approach as one of the position-wise one-to-one distillation methods. This is a grave misunderstanding of our approach and its contributions. In addition, **please note that the “Dimensional Cross-Correlation” matrix shown on the left of Fig.2 has a shape of $D \times D$, where $D$ represents the dimension of BEV features, and each element does not denote the spatial position in the BEV space.**
>
> The core of CD is the **decorrelation mechanism** [R1,R2], which helps to reduce redundancy among dimensional feature components. Consequently, our structural distillation method prevents information collapse in the student model (Tab.3 and Fig.3) and enhances the dimensional feature similarities, as shown in Fig.1-(a) and (b).  It's noteworthy that other reviewers, “4UdP” and “XsGg”, also acknowledged the effectiveness of the CD loss in reducing duplicated information within the student features.
>
> **Temporal Consistency Distillation (TD):**\
> Our baseline student models (BEVFormer [R4], UVTR-CS [R5]) leverage intra-modal temporal features and have shown improved prediction performance compared to single-frame learning. Expanding on the use of temporal information, we propose to further integrate temporal information across different modalities through our Temporal Consistency Distillation (TD). Note that TD exhibits significantly improved prediction performance in our comprehensive component-wise analyses (Tab.1 and 5, Fig.4) as well as comparisons with prior approaches (Tab.4 and 6). To the best of our knowledge, we are the first to offer such new perspectives on temporal cross-modal distillation for 3D object detection, a recognition also shared by other reviewers, “U3eQ”, “X2eU”, and “4Udp”. Hence, we believe that our temporal distillation approach brings a notable contribution to the relevant research fields.
>
> **BEV Feature Shape:**\
> Regarding the modified BEV feature shape parameters (i.e., (200 x 200) $\rightarrow$ (128 x 128)) for BEVFormer [R4], our intention was not to manage GPU memory consumption. Instead, we aimed to maintain consistency with BEVDistill's implementation [R6] for **fair comparisons and to maintain alignment between the feature shape of LiDAR and camera models**.
>
> **Additional Validation on BEVDepth [R11]:**\
> We conducted thorough experiments and comprehensive ablation studies on representative student baselines, including BEVFormer (attention-based model, [R4]) and UVTR (LSS-based model, [R5]), using the nuScenes dataset (see Sec.4). We also compared our method against state-of-the-art distillation methods for 3D object detection. As noted by the majority of other reviewers, these results have extensively and sufficiently validated the effectiveness of our STXD framework. In response to the concerns raised by "1M3u", we extend our experiments to additional baseline student model, BEVDepth [R11], which relies on LSS-based feature generation, depth supervision, and multi-frame inputs. We report results in Tab.R1 of the rebuttal PDF, where our approach outperforms prior distillation methods, such as BEVDistill [R6] and TiG-BEV [R3], by up to +3.2% of NDS. These results are consistent to the results from BEVFormer and UVTR-C/CS baselines (see Tab.4 and 6).
>
> **Comparison with Prior Distillation Approaches:**\
> Until recently, MSE and its variant with focusing on regions with ground-truth (MSE w/ GT) have been commonly adopted in state-of-the-art cross-modal distillation for object detection [R6-R9]. To our best understanding, General Instance Distillation (GID, [R12]) is also conceptually similar to "MSE w/ GT" concerning its sampling strategy around the foreground objects. Based on this observation, we compared our approach with such prevalent methods (MSE, MSE w/ GT) in Tab.2.
> As mentioned above, our correlation regularizing distillation (CD) drew inspiration from self-supervised learning methods [R1,R2]. The lower section of Tab.2. presents comparisons of various learning strategies, including CLIP [R13], VICReg [R2], Barlow-Twins [R1], to validate our design of CD loss (L236-L239).
>
> **Additional Validation on Waymo Datatset:**\
> Generally, the method that performs well on the nuScenes dataset often achieves favorable outcomes in the Waymo dataset, as observed in [R4,R15]. Therefore, we believe our extensive validation and testing results on the nuScenes sufficiently demonstrate the effectiveness of our distillation approach. To accommodate your suggestion, we are conducting experiments on the Waymo dataset, but we cannot report the results from such a large-scale dataset (230K frames over 1TB) within the short rebuttal period. We will report results in the final version of our paper.

---

> > ### Comment · Reviewer_1M3u · 2023-08-20
> > **Official Comments by Reviewer 1M3u**
> >
> > The novelty of 'structural distillation' is limited.  I assure that I've not misunderstand the paper!  The "Dimensional Cross-Correlation" matrix presented in Figure 2 demonstrates a greater weight on the diagonal than it does elsewhere (0.01), thus implying the pivotal role is still allocated to the corresponding position loss, essentially substituting Mean Squared Error (MSE) with a gauge of similarity.
> >
> > The authors not explain the other problems well. 1) For temporal distillation, the author not explain why need the temporal distillation since the BEV feature already obtained the temporal information. 2) The distillation on other dataset is must! Different from the object detection, the task of distillation often fails to generalize across different datasets.  Is there a necessity to modify the distillation structure or adjust the distillation parameters on different datasets?
> >
> > Considering that the author didn't solve my above concerns, I adjusted my score to reject.

---

> > > ### Author Response · Authors · 2023-08-20
> > >
> > > We sincerely appreciate reviewer “1M3u” for taking the time to review our rebuttal. Below we provide clarifications to further address your concerns.
> > >
> > > **MSE vs. CD**\
> > > First of all, we truly understand the possible confusion between MSE and CD (especially, the first term in $L_{CD}$ of Eq.2) as both are used to align the feature-level similarities between different modalities. However, CD considers feature dimensional (''inter-channel'') correlation while MSE aims to minimize position-wise ''global distances'' as shown in the following equations:
> > >
> > > $L_{MSE}=||\mathbf{F}-\mathbf{G}||_2,$
> > >
> > > $L_{CD}=\sum_{i} (1-\mathbf{C}(i,i))^2 + \lambda_{c} \sum_{i} \sum_{j\neq i}\mathbf{C}(i,j)^2, (\text{quoted from Eq.2})$
> > >
> > > where $\mathbf{F} \in \mathbb{R}^{N\times D}$ is the LiDAR BEV features and $\mathbf{G} \in \mathbb{R}^{N\times D}$ is the camera BEV features, $D$ is the channel-wise dimension of BEV features, and $N$ is the number of serialized BEV feature instances.
> > > Thus, $L_{MSE}$ does not have a feature dimensional de/correlation mechanism (the first and the second term of Eq.2) and does not consider information maximization/collapse in cross-modal knowledge distillation (L46-L48, L144-L145). We demonstrated the difference between $L_{MSE}$ and $L_{CD}$ in terms of 3D object detection performance (Tab.2) and quality of distilled information (quantitative results from Tab.2, Tab.3, Fig.3, and qualitative results from Fig.1 and Fig.A3 in the Appendix). Please note that the difference between MSE and CD is also acknowledged by all the other reviewers.
> > >
> > > **Balancing parameter ($\lambda_c$) for off-diagonal term in Eq.2**\
> > > To the best of our understanding, the reviewer “1M3u” considers that the off-diagonal term (the second term in Eq.2) is less weighted because it is multiplied by a relatively small number of $\lambda_c=0.01$. However, this should not be interpreted as if the off-diagonal elements are neglected. The diagonal term and off-diagonal term involve the summation of $D=256$ and  $D(D-1)=65,280$ elements, respectively, where $D$ is the dimension (channel) of features. To balance these, we searched and found $\lambda_c=0.01$ showed the best results in our early design experiments, following the implementation of Barlow Twins [R1].
> > >
> > > **Importance of Temporal Consistency Distillation (TD)**\
> > > As the reviewer “1M3u” noted, we used baseline student models (BEVFormer [R4], UVTR-CS [R5]) that extract and leverage **intra-modal** temporal features (i.e., temporal features from camera-only inputs). However, our motivation was to go beyond utilizing **intra-modal** temporal information alone. Instead, we aimed to further introduce **cross-modal** temporal information from different modalities, such as LiDAR teacher model. Thus we proposed our Temporal Consistency Distillation (TD), which indirectly transfers the teachers’ temporal information to the student model by introducing a temporal similarity map (please refer to Sec.3.3). The sole effect of TD is demonstrated in Tab.1 and 5, where TD consistently improves the performance of baseline student models (up to +1.77% of NDS and +2.15% of mAP). Additionally, the effect of TD is also qualitatively demonstrated in Fig.4, where the student model trained with TD shows improved similarity patterns with the teacher (L299-L314). This highlights that TD can transfer valuable **cross-modal** information beyond the scope of **intra-modal** temporal features.
> > >
> > > **Additional validation of the proposed framework**\
> > > The authors acknowledge that including additional experiments on different datasets would be beneficial. As we mentioned earlier, to accommodate the reviewer's suggestion, we are currently conducting experiments on the Waymo dataset, which will be included in the final version of the paper. As a side note, the official implementations of our baselines, BEVFormer and UVTR, do not immediately support the Waymo dataset. This is another reason why we are unable to conduct additional experiments within the limited rebuttal period.
> > >
> > > However, it must be noted that validating the proposed framework across various teacher-student baselines is also crucial in the field of knowledge distillation. In our experiments, we achieved substantial performance improvements for various baselines on the widely-used nuScenes dataset (Tab.4 and 6), including BEVFormer and UVTR models as well as BEVDepth (Tab.R1 of the rebuttal PDF). Therefore, we believe that our extensive experiments and analyses offer sufficient validation for the proposed STXD framework. More importantly, our framework does not include any specialized designs or parameters specifically tailored to nuScenes dataset. As described in Eq.2 and Eq.4, our structural and temporal distillation losses only require BEV features from those models and do not need modifications of loss functions based on specific datasets.

---

> > > > ### Comment · Reviewer_1M3u · 2023-08-21
> > > > **Additional comments**
> > > >
> > > > 1) Balancing parameter (λc) in Eq.2
> > > >
> > > >     I'm sorry that I misunderstood your implementation here. There is no problem here.
> > > >
> > > > 2) Additional dataset
> > > >
> > > >    BEVformer includes the waymo dataset in its initial release, UVTR not do. As I described before, the distillation task is always different from object detection. The distillation task is more sensitive to the dataset. The BEV distillation is following the distillation of 2D detection, so it’s not a difficult thing to make it work on a specific dataset by adjusting the parameters. A good distillation methods should have good generalization. In fact, the BEVformer has provided the waymo-mini dataset which has 1/5 dataset and easy to train.
> > > >
> > > > Finally, I will change my rating to “bordline reject”, since the method still need more promising contribution and good generalization.

---

> > > > > ### Author Response · Authors · 2023-08-21
> > > > >
> > > > > We appreciate the prompt and considerate response from the reviewer '1M3u.' Below, we provide our answers to address the remaining concerns.
> > > > >
> > > > > **Sensitivity of KD toward different datasets**\
> > > > > We indeed acknowledge that verifying the proposed framework on diverse datasets is valuable. Therefore, in the final version, we will definitely attempt to incorporate the validation results on the Waymo datasets. We also agree with ‘1M3u’ in that the knowledge distillation (KD) introduces a challenging and complex task within conventional object detection (OD) as it aims to transfer valuable information from the teacher to the student model. In our literature review of KD for OD (please refer to L96-L110), we observed that some of the prior works validated their proposed KD approaches across multiple datasets (e.g., 2D OD [15][55] on COCO, Pascal VOC, and CrowdHuman; and 3D OD [18] on KITTI-3D and Waymo). Importantly, these studies conducted evaluations without introducing specific modifications to their distillation methods to accommodate different datasets. Furthermore, certain works demonstrated the adequacy of their KD for OD approaches using a single dataset, as exemplified by MonoDistill [11], LIGA-Stereo [16], and UVTR [27]. From our understanding, the majority of previous studies on KD for 2D/3D OD have primarily focused on carefully aligning and matching the feature distributions between the teacher and the student models [9,10,11,15,16,18,27,55], rather than investigating the sensitivity of KD approaches to different datasets. We would be greatly appreciated if the reviewer '1M3u' could offer guidance regarding the particular aspects of datasets that play a critical and sensitive role in the application of KD for OD approaches. This understanding seems to be crucial for us to better understand the argument of ‘1M3u’ concerning “distillation task is more sensitive to the dataset”.
> > > > >
> > > > > References from the manuscript:\
> > > > > [9] Chen et al. (2023). BEVDistill: Cross-modal bev distillation for multi-view 3d object detection. The Eleventh International Conference on Learning Representations. 2022.
> > > > >
> > > > > [10] Chen et al. (2021). Disentangle your dense object detector. In Proceedings of the 29th ACM international conference on multimedia (pp. 4939-4948).
> > > > >
> > > > > [11] Chong et al. (2022). MonoDistill: Learning Spatial Features for Monocular 3D Object Detection. In International Conference on Learning Representations.
> > > > >
> > > > > [15] Guo et al. (2021). Distilling object detectors via decoupled features. In Proceedings of the IEEE/CVF Conference on Computer Vision and Pattern Recognition (pp. 2154-2164).
> > > > >
> > > > > [16] Guo et al. (2021). Liga-stereo: Learning lidar geometry aware representations for stereo-based 3d detector. In Proceedings of the IEEE/CVF International Conference on Computer Vision (pp. 3153-3163).
> > > > >
> > > > > [18] Hong et al. (2022). Cross-modality knowledge distillation network for monocular 3d object detection. In European Conference on Computer Vision (pp. 87-104). Cham: Springer Nature Switzerland.
> > > > >
> > > > > [27] Li et al. (2022). Unifying voxel-based representation with transformer for 3d object detection. Advances in Neural Information Processing Systems, 35, 18442-18455.
> > > > >
> > > > > [55] Yang et al. (2022). Prediction-guided distillation for dense object detection. In European Conference on Computer Vision (pp. 123-138). Cham: Springer Nature Switzerland.
> > > > >
> > > > > **Challenges in implementing Waymo dataset**\
> > > > > In response to the reviewer's suggestion, we have attempted to validate our framework using the Waymo dataset. However, it is considerably challenging to find and implement available codes for the Waymo dataset and obtain results within this short rebuttal period.
> > > > >
> > > > > As discussed in the issue page ([link](https://github.com/fundamentalvision/BEVFormer/issues/34)), the official git repository of BEVFormer ([link](https://github.com/fundamentalvision/BEVFormer)) currently does not provide training codes for the Waymo dataset.
> > > > > Following the issue ([link](https://github.com/fundamentalvision/BEVFormer/issues/112)), we also found another repository ([link](https://github.com/OpenDriveLab/Birds-eye-view-Perception)), which contains code for the Waymo mini dataset similar to what ‘1M3u’ mentioned.
> > > > > Unfortunately, this repository no longer provides any downloadable links for the Waymo mini dataset.(i.e., in the data preparation guide ([link](https://github.com/OpenDriveLab/Birds-eye-view-Perception/blob/master/docs/data_preparation.md)), “Google Drive” link does not work.)
> > > > > Instead, we attempted an alternative approach to parsing the Waymo dataset. However, due to the absence of necessary files included in the unavailable “Google Drive” link, we still face challenges in conducting an experiment on the Waymo dataset.
> > > > >
> > > > > Due to these challenges, it is considerably difficult to provide a comprehensive evaluation of the Waymo dataset within the short rebuttal period. In the final version, we will definitely attempt to include the validation results on the Waymo datasets.

---

### Official Review · Reviewer_4Udp · 2023-07-06

**Soundness:** 4 excellent
**Presentation:** 4 excellent
**Contribution:** 4 excellent
**Rating:** 7
**Confidence:** 4

**Summary:**

The authors propose a new method for LiDAR-to-Camera 3D object detection model distillation. They propose distillation along three avenues. First, referencing Barlow Twins, they consider redundancy reduction through cross-correlation between LiDAR and Camera features. Second, they enforce consistency between temporal similarity maps of current lidar-past lidar and current camera-past lidar. Finally, they additionally have distillation at the output level.

**Strengths:**

- The paper is clear and easy to read, with informative diagrams.
- The motivation for the distillation approaches is sound. I especially appreciate the temporal similarity distillation, as well as its improvement on mAVE.
- Step-by-step ablations demonstrate the improvement from each proposed component.
- Analyses in pg 7 demonstrate the redundancy reduction of the core distillation method.


**Weaknesses:**

- Additional ablations on Temporal Consistency Distillation would be helpful. For instance, perhaps a more natural way to do this could be to generate temporal similarity maps independently for LiDAR and Camera (current lidar x past lidar, current camera x past camera), and enforce consistency between those instead. It is not entirely clear why current camera x past lidar is done instead.
- Discussion and head-to-head comparison with TiG-BEV with settings shown in the TiG-BEV paper is strongly recommended.
- As the proposed method considers inter-channel relationships, discussion of the proposed method’s relationship to works that use inter-channel distillation (such as in TiG-BEV) would be appreciated.
- How are teacher and student outputs matched for the response distillation? Is it greedy, hungarian, or something else?


**Questions:**

- L138: Why is N = X*Y*Z? It instead seemed like N should be X*Y, the flattened BEV feature map. Further, is this done over all the samples in a batch or separately for each BEV teacher-student pair?

**Limitations:**

Discussed

---

> ### Author Rebuttal · Authors · 2023-08-09
>
> We express our gratitude to the reviewer "4Udp" for recognizing our fundamental contributions to feature-level distillation approaches, particularly highlighting the significance of Temporal Consistency Distillation, and acknowledging our thorough validation results on a component-wise basis. Below, we provide answers to your questions and hope they appear to be reasonable as well.
>
> **Design Rationale behind Temporal Consistency Distillation (TD):**\
> "4Udp" suggested an ablation of different strategies for temporal similarity map generation, specifically focusing on modality-independent associations like "current LiDAR ($F_0$) x past LiDAR ($F_k$), current camera ($G_0$) x past camera ($G_k$)". In our early design experiments, we also tested the same similarity map generation strategy (i.e., $L_{TD}=D_{KL}(S || T)$ where $S=G_0 G_k^{T}, T=F_0 F_k^{T}$). However, this formulation did not result in a significant improvement in detection performance as we had anticipated. Please note that, in our cross-modal knowledge distillation setting, the LiDAR teacher model can provide informative and high-quality features across temporal frames, leveraging its pre-trained weights. However, the image features from past frames, generated via the student model, offer relatively unreliable and uninformative guidance compared to the teacher model. Consequently, relying on these untrustworthy past image features may even hinder the progressive learning of informative features from the teacher model. Thus, we devised the current formulation of $L_{TD}$, where the LiDAR teacher is primarily utilized to generate features from past frames and guide the student model, as presented in Eq.3 and 4. Consequently, TD led to significantly improved prediction performance in our component-wise analyses, as shown in Tab.1.
>
> **Comparison with TiG-BEV [R3]:**\
> As pointed out by "4Udp", our cross-correlation regularizing distillation (CD) and TiG-BEV both take into consideration dimensional (inter-channel) correlation. However, we want to emphasize that these losses are distinct in terms of their underlying objectives and specific implementations.
>
> According to our understanding of TiG-BEV, the inter-channel BEV distillation loss ("IC loss") encourages the student's features to replicate the dimensional **self-correlation** ($D \times D$ matrix, $D$=the dimension of BEV features) of the teacher's features. However, IC loss does not consider cross-modal associations among BEV features from the teacher and student models.
>
> In contrast, the core contribution and novelty of CD is introducing **cross-modal cross-correlation regularization** to prevent student features from being redundant. Inspired by recent info-max oriented self-supervised learning approaches (e.g., Barlow Twins [R1] and VICReg [R2]), CD exploits the **decorrelation mechanism** (the second term of Eq.2) which prevents information collapse in the student model. CD also enhances the dimensional component-wise feature similarities (the first term of Eq.2). Please refer to L43-L48 and L142-L145 for detailed explanations. As "4Udp" also remarked, our extensive analyses presented in Tab.3 and Fig.3 prove the effectiveness of CD in terms of eliminating redundant information in the features of the student, and ultimately contributes to significantly improved prediction performance. However, TiG-BEV does not have such information maximization mechanism, and only relies on the global one-to-one MSE distance (L2-norm) between modality-specific self-correlation matrices. Hence, CD pursues a different learning principle compared to IC loss of TiG-BEV.
>
> For a clear comparison, we also have illustrated the differences in loss calculation in Fig.R2 of the rebuttal PDF. Again, one notable distinction is that TiG-BEV computes self-correlation for each modality separately (Fig.R2-(b)), while our CD loss calculates cross-correlation between LiDAR and camera features (Fig.R2-(a)).
>
> Last but not least, we provide an additional quantitative comparison with TiG-BEV using BEVDepth [R11] as a baseline student model. Tab.R1 of the rebuttal PDF shows that both CD and TD outperforms TiG-BEV (correlation-based distillation losses, FD), by achieving notable improvements of +3.2% and +3.1% in terms of NDS, respectively. It is noteworthy that CD achieves superior performance, even when compared with TiG-BEV leveraging additional inner-depth supervision (+2.3% of NDS). In addition, we also measure the effective dimension $d_{\text{eff}}$ and dimensional redundancy to compare the redundancy in student features learned from the teacher in Tab.R2. These results consistently demonstrate the significance of mitigating feature redundancy in cross-modal knowledge distillation and enhancing feature quality through our feature-level distillation approaches. We will include these additional results in the final version of the Appendix.
>
> **Further Clarifications**
> - We adopted the commonly used Hungarian algorithm [R14] to define set-to-set matching between the ground-truth and candidates from the teacher and the student, separately. Then, we constructed a mapping function (i.e., $\pi(j)$ in Eq.6) between candidates from the teacher and the student based on matched ground-truth indices. We will better explain this in Sec.3.4 in the final version.
> - We define a shape of BEV feature as $N = X \cdot Y \cdot Z$, using a generalized notation where $Z=1$ for BEVFormer [R4] and $Z=11$ for UVTR-C/CS [R5], following the official implementation of each method (see Sec.A.1 of the Appendix). Also we did not combine the BEV teacher-student pairs across batches; instead, we handled them separately.

---

> > ### Comment · Reviewer_4Udp · 2023-08-20
> >
> > I appreciate the detailed response from the authors. It is intuitive to me now that past lidar teacher frames are better to learn from compared to past camera features. I also find the additional experiments in Table R1 convincing.
> >
> > As such, I maintain my original rating.

---

### Author Rebuttal · Authors · 2023-08-09

Dear AC and Reviewers,

First of all, we sincerely appreciate your time and effort to review our work. At the end of this comment, we have attached our **Rebuttal PDF**, which includes additional experimental results and diagrams illustrating the comparisons with prior methods. Tables and figures in the rebuttal PDF are referenced in our rebuttal comments as Tab.**R**x and Fig.**R**x. We will include these additional experimental results (Tab.R1 and Tab.R2) in the final version. Also, we have listed the references we used in our rebuttal below. Please refer to this list in each rebuttal comment and in the rebuttal PDF.

**References:**

**[R1]** Zbontar, J., Jing, L., Misra, I., LeCun, Y., & Deny, S. (2021, July). Barlow twins: Self-supervised learning via redundancy reduction. In International Conference on Machine Learning (pp. 12310-12320). PMLR.

**[R2]** Bardes, A., Ponce, J., & Lecun, Y. (2022, April). VICReg: Variance-Invariance-Covariance Regularization For Self-Supervised Learning. In ICLR 2022-International Conference on Learning Representations.

**[R3]** Huang, P., Liu, L., Zhang, R., Zhang, S., Xu, X., Wang, B., & Liu, G. (2022). TiG-BEV: Multi-view bev 3d object detection via target inner-geometry learning. arXiv preprint arXiv:2212.13979.

**[R4]** Li, Z., Wang, W., Li, H., Xie, E., Sima, C., Lu, T., ... & Dai, J. (2022, October). BEVFormer: Learning bird’s-eye-view representation from multi-camera images via spatiotemporal transformers. In European conference on computer vision (pp. 1-18). Cham: Springer Nature Switzerland.

**[R5]** Li, Y., Chen, Y., Qi, X., Li, Z., Sun, J., & Jia, J. (2022). Unifying voxel-based representation with transformer for 3d object detection. Advances in Neural Information Processing Systems, 35, 18442-18455.

**[R6]** Chen, Z., Li, Z., Zhang, S., Fang, L., Jiang, Q., & Zhao, F. (2022). BEVDistill: Cross-modal bev distillation for multi-view 3d object detection. arXiv preprint arXiv:2211.09386.

**[R7]** Chen, Z., Yang, C., Li, Q., Zhao, F., Zha, Z. J., & Wu, F. (2021, October). Disentangle your dense object detector. In Proceedings of the 29th ACM international conference on multimedia (pp. 4939-4948).

**[R8]** Hong, Y., Dai, H., & Ding, Y. (2022, October). Cross-modality knowledge distillation network for monocular 3d object detection. In European Conference on Computer Vision (pp. 87-104). Cham: Springer Nature Switzerland.

**[R9]** Yang, C., Ochal, M., Storkey, A., & Crowley, E. J. (2022, October). Prediction-guided distillation for dense object detection. In European Conference on Computer Vision (pp. 123-138). Cham: Springer Nature Switzerland.

**[R10]** Zhu, B., Jiang, Z., Zhou, X., Li, Z., & Yu, G. (2019). Class-balanced grouping and sampling for point cloud 3d object detection. arXiv preprint arXiv:1908.09492.

**[R11]** Li, Y., Ge, Z., Yu, G., Yang, J., Wang, Z., Shi, Y., ... & Li, Z. (2023, June). BEVDepth: Acquisition of reliable depth for multi-view 3d object detection. In Proceedings of the AAAI Conference on Artificial Intelligence (Vol. 37, No. 2, pp. 1477-1485).

**[R12]** Dai, X., Jiang, Z., Wu, Z., Bao, Y., Wang, Z., Liu, S., & Zhou, E. (2021). General instance distillation for object detection. In Proceedings of the IEEE/CVF conference on computer vision and pattern recognition (pp. 7842-7851).

**[R13]** Radford, A., Kim, J. W., Hallacy, C., Ramesh, A., Goh, G., Agarwal, S., ... & Sutskever, I. (2021, July). Learning transferable visual models from natural language supervision. In International conference on machine learning (pp. 8748-8763). PMLR.

**[R14]** Kuhn, H. W. (2005). The Hungarian method for the assignment problem. Naval Research Logistics (NRL), 52(1), 7-21.

**[R15]** Yin, T., Zhou, X., & Krahenbuhl, P. (2021). Center-based 3d object detection and tracking. In Proceedings of the IEEE/CVF conference on computer vision and pattern recognition (pp. 11784-11793).

---

### Decision · Program_Chairs · 2023-09-21

**Decision:**

Accept (poster)

**Comment:**

This paper presented a novel approach for LiDAR-to-Camera distillation. Three key schemes are introduced, and experimental results demonstrate the proposed approach. The reviewers provided many constructive comments. Most comments were well handled by the author rebuttal. There remain two key issues: add experiments with longer past frames and on an additional dataset. Please do include the two experiments in the final version.